# Current glacier recession causes significant rockfall increase:
# The immediate paraglacial response of deglaciating cirque walls

Ingo Hartmeyer[1], Robert Delleske[1], Markus Keuschnig[1], Michael Krautblatter[2], Andreas Lang[3], Lothar Schrott[4], Jan-Christoph Otto[3]

[1]GEORESEARCH Research Institute, Wals, 5071, Austria
[2]Chair of Landslide Research, Technical University of Munich, Munich, 80333, Germany
[3]Department of Geography and Geology, University of Salzburg, Salzburg, 5020, Austria
[4]Department of Geography, University of Bonn, Bonn, 53115, Germany

*Correspondence to*: Ingo Hartmeyer (ingo.hartmeyer@georesearch.ac.at)

**Abstract.** In the European Alps almost half the glacier volume disappeared over the past 150 years. The loss is reflected in glacier retreat and ice surface lowering even at high altitude. In steep glacial cirques surface lowering exposes rock to atmospheric conditions probably for the very first time in several millennia. Instability of rockwalls has long been identified as one of the direct consequences of deglaciation, but so far cirque-wide quantification of rockfall at high-resolution is missing. Based on terrestrial LiDAR a rockfall inventory for the permafrost-affected rockwalls of two rapidly deglaciating cirques in the Central Alps of Austria (Kitzsteinhorn) is established. Over six-years (2011-2017) 78 rockwall scans were acquired to generate data of high spatial and temporal resolution. 632 rockfalls were registered ranging from 0.003 to 879.4 m³, mainly originating from pre-existing structural rock weaknesses. 60 % of the rockfall volume detached from less than ten vertical meters above the glacier surface, indicating enhanced rockfall activity over tens of years following deglaciation. Debuttressing seems to play a minor effect only. Rather, preconditioning is assumed to start inside the Randkluft (void between cirque wall and glacier) where measured sustained freezing and ample supply of liquid water likely cause enhanced physical weathering and high quarrying stresses. Following deglaciation, pronounced thermomechanical strain is induced and an active layer penetrates into the formerly perennially frozen bedrock. These factors likely cause the observed paraglacial rockfall increase close to the glacier surface. This paper, the first of two companion pieces, presents the most extensive dataset of high-alpine rockfall to date and the first systematic documentation of a cirque-wide erosion response of glaciated rockwalls to recent climate warming.

## 1 Introduction

High-alpine, glacial environments are severely affected by recent climate warming (WGMS, 2017). This is especially true for the European Alps, where mean temperature rise over the last 150 years more than doubled the global mean (Böhm, 2012) and over this period approximately 50 % of the glacier volume has disappeared (Haeberli et al., 2007). Glacier retreat rates increased since the 1980s and have been exceeding historical precedents in the early 21st century (Zemp et al., 2015). The

consequences of these changes are most visible in lower lying glacierized cirques where ice-surface lowering in the ablation area is particularly apparent (Kaser et al., 2006; Pelto, 2010) and exposes cirque walls to the atmosphere probably for the first time in several millennia. Paleoclimatic studies assume that glaciers are currently shrinking to extents that are unprecedented since the Medieval (~ 1 ka ago) or Roman Warm Periods (~ 2 ka ago) (Holzhauser et al., 2005; Joerin et al., 2006), or more

likely, since the mid-Holocene Warm Period (~ 5 ka ago) (Joerin et al., 2008; Auer et al., 2014; Solomina et al., 2015).

Rockwall characteristics strongly depend on preconditioning stress fields (Krautblatter and Moore, 2014). Especially parameters such as fracture density and orientation are first order controls on rock slope erosion (Sass, 2005; Moore et al., 2009). Glacial oversteepening increases the stress regime acting within cirque walls and promotes rock slope failures at various scales (Ballantyne, 2002; de Haas et al., 2015). Ice surface lowering alters ground thermal conditions (Wegmann et al., 1998),

modifies pre-existing slope stresses (Augustinus, 1995; Leith et al., 2014), and therefore potentially causes local instability and elevated mass wasting activity. This has significant implications for risk management in high-alpine environments – especially when considering the growing popularity of glacier tourism (Fischer et al., 2011a; Purdie, 2013). Steep gradients (due to glacial oversteepening) in the surrounding rockwalls and the low friction on the glacier surface both promote long rockfall runouts underneath cirque walls (Schober et al., 2012) putting nearby infrastructure at risk. Continued climate warming

is expected to exacerbate this issue, making long-term rockwall monitoring an essential prerequisite for rockfall risk assessment in glacial environments (Stoffel and Huggel, 2012).

Frost action is considered a key agent in preparing and triggering high-alpine rockfall (Draebing and Krautblatter, 2019) and a major driver of rock slope erosion in cold environments (e.g. Hales and Roering, 2009). Only recently a number of studies demonstrated cirque wall retreat rates exceeding rates of glacial incision, underlining the contribution of frost weathering to

the shaping of 'glacial' landscapes (Oskin and Burbank, 2005; Naylor and Gabet, 2007; Scherler et al, 2011). Frost weathering processes encompass volumetric ice expansion and ice segregation which are theoretically able to produce pressures exceeding the tensile strength of rocks (Hallet et al., 1991; Matsuoka and Murton, 2008). Volumetric expansion results from freezing of in-situ water and requires high water saturation and extreme cooling rates (Walder and Hallet, 1986; Matsuoka and Murton, 2008). Ice segregation causes cryosuction-induced migration of unfrozen water toward freezing fronts (Walder and Hallet,

1985) and is effective in hard, low-porosity rock at a wide range of sustained sub-zero temperatures (Girard et al., 2013; Duca et al., 2014; Murton et al., 2016). Recent lab studies highlight the importance of fatigue damage under different frost weathering regimes and in different rock types and indicate that subcritical crack propagation plays a key role in the generation of rockfalls in periglacial environments (Jia et al., 2015, 2017).

Rockfall or rock slope failures that are spatiotemporally related to the transition from glacial conditions to non-glacial

conditions have been termed 'paraglacial' (McColl, 2012). The paraglacial concept incorporates processes, materials and landforms that are directly conditioned by former glaciation and deglaciation (Church and Ryder, 1972; Ballantyne, 2002). Studies on paraglacial rock slope readjustment often focus on enhanced rates of geomorphic activity after/during deglaciation mainly on rare high-magnitude slope failures. Frequent low-magnitude failure patterns have received comparably little attention. Numerous studies on paraglacial bedrock erosion have focused on Late Pleistocene to Holocene timescales that

relate to glacier retreat from Last Glacial maximum (LGM) positions. Relevant studies include extensive mapping of slope instabilities (Allen et al., 2010), terrestrial cosmogenic nuclide dating of post-glacial rock slope failures (Cossart et al. 2008; Ballantyne et al., 2014), effects of glacial debuttressing (McColl and Davies, 2012) and numerical modelling of fracture initiation and propagation during glacial (un)loading (Grämiger et al., 2017).

On a more recent timescale the effects of glacier shrinkage from Little Ice Age (LIA) limits and increased mass wasting activity are unravelled using field mapping (Deline, 2009), photo comparisons (Ravanel and Deline, 2010), GIS analyses (Holm et al., 2004), and historical documentation (Noetzli et al., 2003). Paraglacial adjustment to the most recent episode of glacial recession – i.e. the dramatic glacier retreat observed over the past few decades – has so far only marginally been addressed. In the Alps, singular, high-magnitude events were examined in the Mont Blanc Massif, France (Deline et al., 2008), at the Piz Kesch, Switzerland (Phillips et al., 2017), and adjacent to the Aletsch Glacier, Switzerland (Manconi et al., 2018) and have at least partially been attributed to current glacier melting. Quantitative studies of lower magnitude paraglacial rockfalls are rare and include a detailed topographic study of rock and ice avalanches in the Monte Rosa east-face, Italy (Fischer et al. 2011b), a four-year time series on a paragneiss ridge at the Gemsstock ski area, Switzerland (Kenner et al., 2011), a two-year monitoring from the Tour Ronde east-face, France (Rabatel et al., 2008), and slope stability surveys from the surroundings of the Refuge des Cosmiques, France (Ravanel et al., 2013).

Quantification of paraglacial rockfall release over larger surfaces and over several years is missing – in large parts due to the harsh, high-alpine environmental conditions – and effectively hinders evaluating the impacts of current glacier retreat on rockfall occurrence. Using data from a six-year terrestrial LiDAR monitoring campaign (2011-2017), we present a rockfall inventory from the Central Alps of Austria that is unique for high-alpine study areas in spatial and temporal extent, and level of detail. We (i) systematically quantify rockfall in two neighbouring, glacial cirques, (ii) reveal significantly increased (paraglacial) rockfall in recently deglaciated rockwall sections immediately above the current glacier surface, (iii) use a unique multiyear set of bedrock temperatures acquired inside the Randkluft to quantify thermal effects of deglaciation on adjacent rockwalls, and (iv) identify antecedent rockfall preparation inside the Randkluft (subcritical crack propagation driven by ice segregation, quarrying-related tensile stress) and subsequent deglaciation-induced thermal forcing as most likely causes for the observed glacier-proximal concentration of rockfall source areas.

Here, after documenting study area and method applied, an inventory of mass movements is presented. Data quality is analysed, spatial patterns of rockfall and rockfall failure depth are presented and causes of the observed rockfall patterns discussed. Magnitude-frequency relationships and rockwall retreat rates derived from this data are discussed in a companion study (Hartmeyer et al., 2020).

## 2 Study area

Two cirques located in the summit region of the Kitzsteinhorn (3,203 m above sea level), Hohe Tauern Range, Austria (Fig. 1), immediately northwest of the summit were selected for monitoring. Both cirques are occupied by the Schmiedingerkees, a

glacier which is home to Austria's oldest glacier ski-area. Since 2010 an extensive, multi-scale monitoring of permafrost-rockfall interaction ('Open-Air-Lab Kitzsteinhorn') (Keuschnig et al., 2015) includes several deep and shallow boreholes (Hartmeyer et al., 2012), two permanently installed electrical resistivity tomography profiles (Supper et al., 2014; Keuschnig et al., 2016), rock anchor load loggers (Plaesken et al., 2017), extensometers in fractures (Ewald et al., 2019), and several fully automated weather stations.

All rockwalls investigated here tower above the Schmiedingerkees: the Kitzsteinhorn north-face (KN), the Kitzsteinhorn northwest-face (KNW), the Magnetkoepfl east-face (MKE), the Magnetkoepfl west-face (MKW) and the Maurergrat east-face (MGE) (Figs. 1, 2). The total surface area of all rockwalls studied is 234,700 m² and with an area of 133,400 m² and a mean height of roughly 200 m KNW is the largest rockwall studied. Slope gradients increase towards the glacier surface, as is characteristic for cirque walls worldwide (Sanders et al., 2012). With 72° the steepest mean gradient occurs at MKE, followed by MKW (63°), and KNW displays the lowest gradient (44°) (Table S1).

The investigated rockwalls developed in rocks of the Glockner Nappe, mainly calcareous micaschists with isolated occurrences of marble and serpentinite especially at the Magnetkoepfl (Cornelius and Clar, 1935; Hoeck et al., 1994). Cleavage orientation in the predominant calcareous micaschists is similar at all rockwalls studied and dips steeply (~ 45°) to NNE (Fig. 3). Numerous pronounced joint-sets indicate high degrees of fracturing, which is particularly evident along existing tectonic faults (e.g. at KNW) and along distinct cleavage planes (e.g. at KN, MKE). The two most prominent joint sets dip subvertical to W (J1) and steeply to SW (J2). Investigations of rock mass strength carried out in all investigated rockwalls indicate highly variable lithologic strength due to the high spatial variability in fracture density (Terweh, 2012).

Mean temperature during the study period (2011-2017) recorded at a weather station located on the Schmiedingerkees (Fig. 1) was -2.0 °C. According to an empirical-statistical model of permafrost distribution for the Hohe Tauern range (Schrott et al., 2012) permafrost can be expected above 2,500 m a.s.l. on north-facing slopes and above 3,000 m a.s.l. on south-facing slopes.

## 2.1 Recent glacier shrinkage

The Schmiedingerkees has retreated considerably in recent decades and ice-aprons have degraded significantly in the surrounding cirque walls (Figs. 1, 2, 4). The oldest useable air photos date back to 1953 (Land Salzburg, 1953) and demonstrate a glacier area of 3.2 km². Since then the Schmiedingerkees lost more than half of its size (-56 %) and adjacent to the monitored rockwalls thinned by an average of 17 m (Fig. 2 ,Table S2). Long-term glaciological monitoring at the nearby glacier Stubacher Sonnblickkees (located 9 km SW of the study area) shows evened mass balances between the 1950s and the early 1980s (Slupetzky and Ehgartner, 2014; Slupetzky, 2015) indicating that most of this surface change has occurred over the last 35 years.

In 2008 the first comprehensive terrain data was acquired for the Schmiedingerkees (Land Salzburg, 2008) using airborne laserscanning. The comparison with current UAV-derived terrain data demonstrates that in the period between 2008 and 2017 glacier volume decreased by 8.5 $10^6$ m³. Mass loss was most pronounced near the terminus, but also in the accumulation area, i.e. adjacent to the rockwalls in focus here, distinct ice-apron degradation and glacier retreat is evident. Mean ice surface

lowering next to the monitored rockwalls equalled 0.7 m a$^{-1}$ over the study period and exposed large, fresh bedrock surfaces (Fig. 1, Table S2).

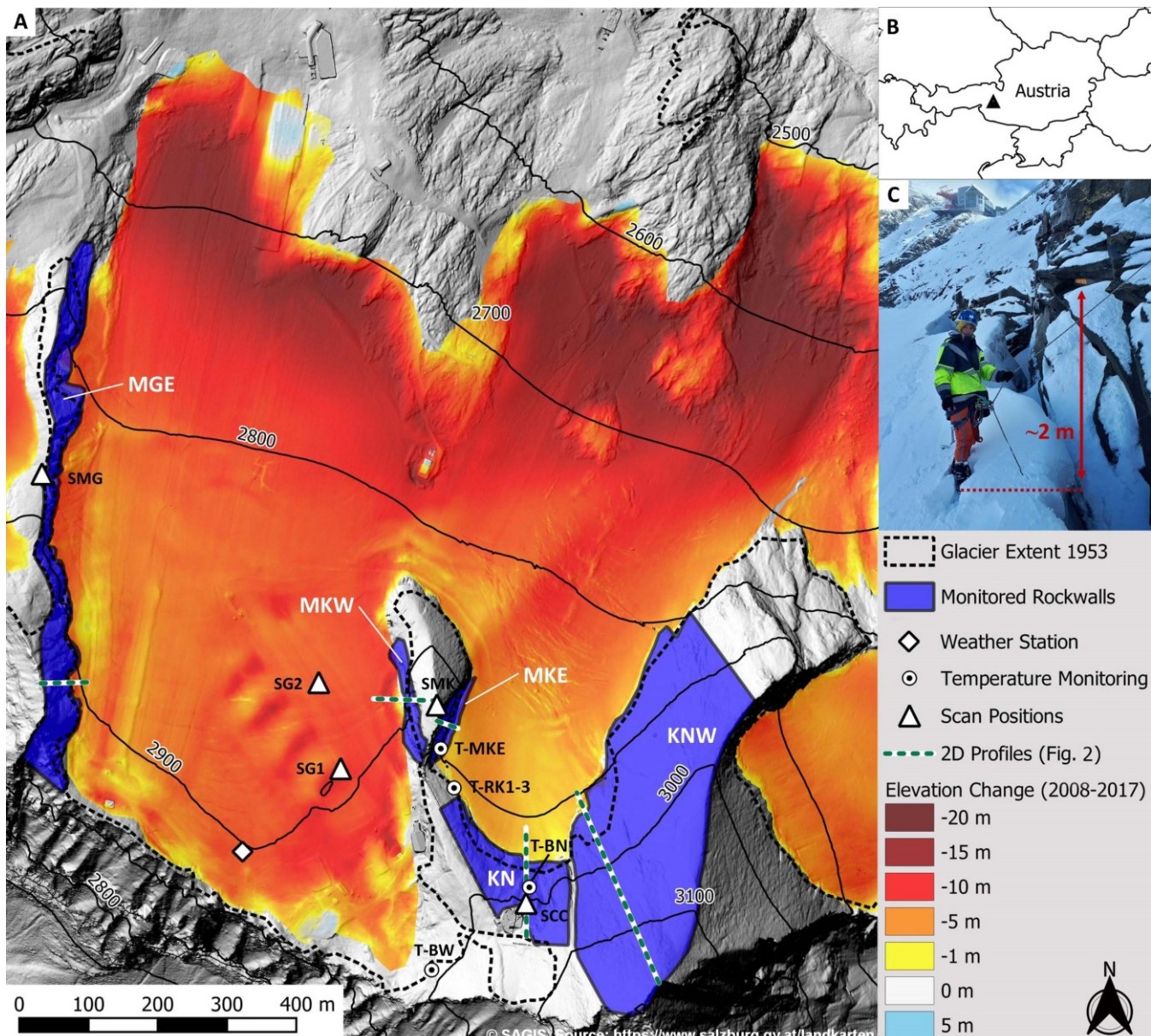

**Figure 1A: Hillshade of study area with monitored rockwalls, scan positions, 1953 glacier extent, and elevation changes of the surface of the Schmiedingerkees between 2008 and 2017. While glacial thinning is most evident near the terminus, pronounced ice surface**
**lowering (~ 0.7 m a$^{-1}$) is also observed adjacent to the monitored cirque walls. 1B: Location of study site within Austria (47.19 °N, 12.68 °E). 1C: Glacier surface lowering 2015-2019. Orange spray marker on rockwall indicates glacier surface level on 04.09.2015 (photo: I. Hartmeyer, 09.09.2019). Abbreviations: KN = Kitzsteinhorn north-face, KNW = Kitzsteinhorn northwest-face, MKE = Magnetkoepfl east-face, MKW = Magnetkoepfl west-face, MGE = Maurergrat east-face, SMK = Scan Position 'Magnetkoepfl', SCC = Scan Position 'Cable Car Top Station', SG1 = Scan Position 'Glacier 1', SG2 = Scan Position 'Glacier 2', SMG = Scan Position**
**'Maurergrat', T-BN = Deep borehole (30 m) at Kitzsteinhorn north-face, T-BW = Deep borehole (30 m) at Kitzsteinhorn west-face, T-MKE = Shallow borehole (0.8 m) at Magnetkoepfl east-face, T-RK1-3 = Shallow boreholes (0.8 m) inside Randkluft.**

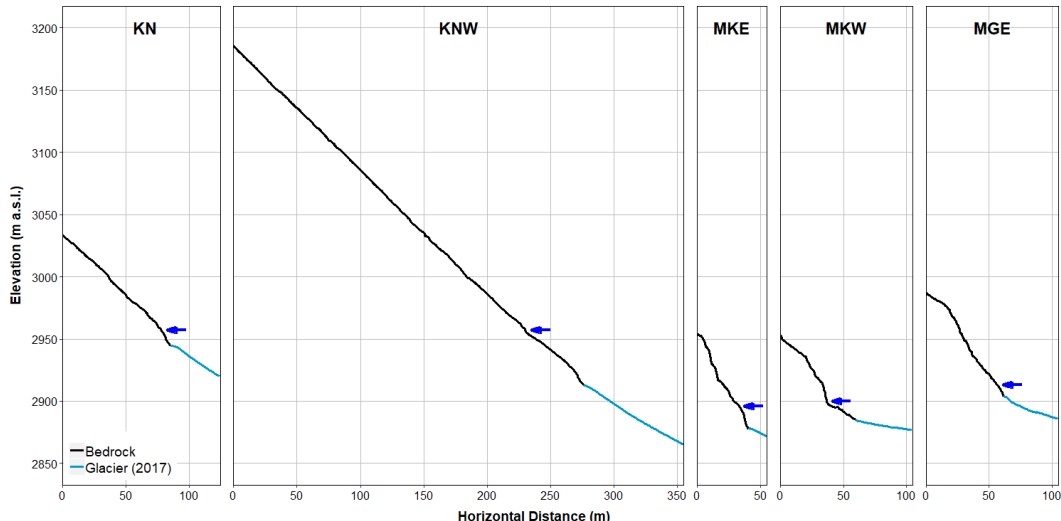

**Figure 2: 2D slope profiles for all monitored rockwalls. Blue arrows indicate the approximate level of the glacier surface in 1953.**

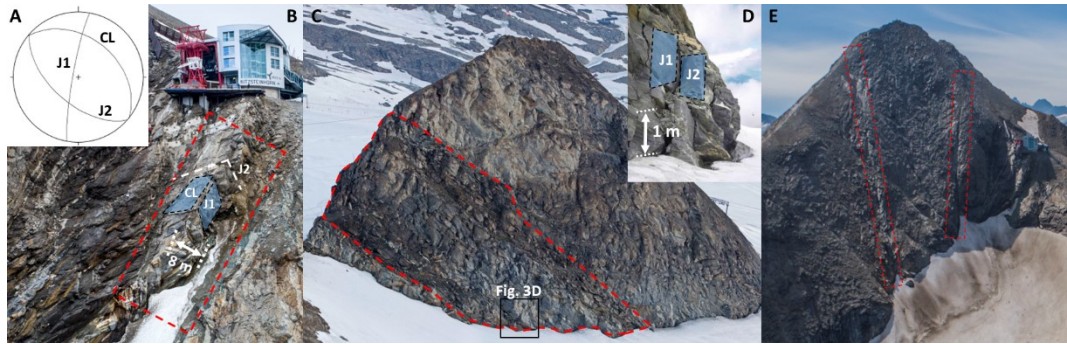

Figure 3: Geological structure and weakness zones at the monitored rockwalls. 3A: Cleavage (CL) of the calcareous mica-schists dips about 45° NNE. Joint sets J1 (dipping subvertical to W) and J2 (dipping steeply to SW) are approximately orthogonal to CL and predispose north-facing slopes for dip-slope failures; 3B: Highly fractured, slope-parallel escarpment at KN (photo: R. Delleske, 01.08.2018); 3C: Diagonal weakness zone following the direction of cleavage at MKE (photo: R. Delleske, 18.07.2014); 3D: Steep joint sets (J1, J2) predispose east- and west-facing areas to toppling failures (photo: A. Schober, 28.07.2010); 3E: Prominent fault lines resulting from ductile shearing at KNW (photo: R. Delleske, 27.08.2019).

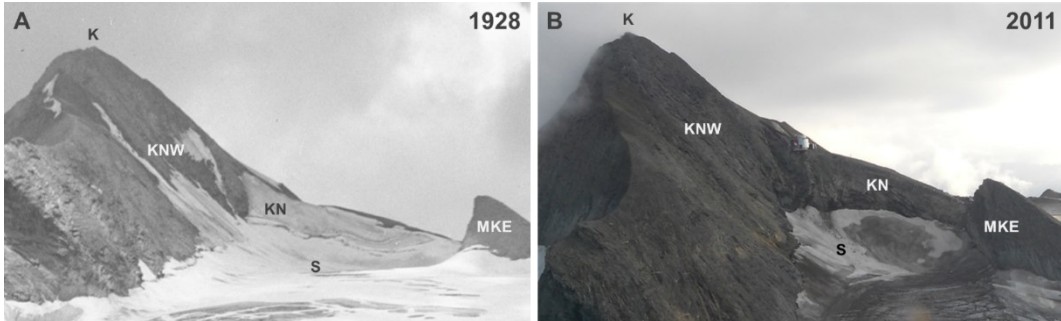

Figure 4: View of Kitzsteinhorn (K) (3,203 m a.s.l.) and Schmiedingerkees (S). 4A: September 1928 (Photo: Stadtarchiv Salzburg, Fotosammlung Josef Kettenhuemer). 4B: September 2011 (Photo: Heinz Kugler). During the reference period the ice surface has lowered considerably while all ice-aprons have completely disappeared. Much of the surface change has occurred since the 1980s (Slupetzky, 2015). Abbreviations: K = Kitzsteinhorn summit, KN = Kitzsteinhorn north-face, KNW = Kitzsteinhorn northwest-face, MKE = Magnetkoepfl east-face, S = Schmiedingerkees.

## 3 Methods

### 3.1 Terrestrial LiDAR monitoring

#### 3.1.1 Data acquisition

Terrestrial LiDAR data acquisition was performed using a Riegl LMS-Z620i laserscanner. A calibrated high-resolution digital camera was mounted on the laserscanner for capturing referenced colour images. Reflectivity on bedrock surfaces was excellent in the near-infrared wavelength used by the scanner, while reflectivity on fresh snow or ice was poor and returned little or no data. Reflectors were not used during data acquisition due to considerable rockfall hazard in the steep, unstable rockwalls.

First LiDAR data was acquired in July/August 2011 at all monitored rockwalls except MKW where data acquisition started in 2012. Data acquisition was restricted to the summer season (May to October). In total 78 rockwall scans were carried out from five different scan positions (Fig. 1). Of these, 22 scans were excluded from further analyses due to snow cover. Scan position 'Maurergrat' was abandoned in 2016, as due to continued glacial thinning site access was lost. Rockwall scans were repeated several times per summer season and at least once per season towards the end of the ablation period. The last scan of all 170 rockwalls was carried out in August 2017, except for MKW that was excluded from further analysis, as unstable blocks were cleared away earlier in 2017 to reduce hazards for a new lift track.

The mean object distances (i.e. distance between scanner and rockwall) differed considerably, varying between 140 m for MKW and 650 m for MGE. The acquisition resolution ranged typically between 0.01-0.02°, resulting in a point cloud resolution mostly between 0.1-0.3 m (see Table S3 for full list of data acquisition parameters).

#### 3.1.2 Data analysis

Airborne LiDAR datasets acquired in 2008 (Land Salzburg, 2008) were used as base data for georeferencing. Alignment of the acquired sequential point clouds was performed based on surface geometry matching within RiScanPro 1.8. First, point clouds were coarsely registered using the GPS location of the scan position and the azimuth angle of the laserscanner. Numerous techniques exist for the fine registration of point clouds, which include the Iterative-Closest-Point (ICP) algorithm 180 (Besl and McKay, 1992; Chen and Medioni, 1992), 3D Least Squares Matching (Akca, 2007), point-to-plane approaches (Grant et al., 2012). Here we used the ICP-algorithm, a cloud matching technique for finding the transformation between two point clouds by minimizing the square errors between corresponding entities. Consistent with previous studies on rock slope systems (Rosser et al., 2007; Abellán et al., 2011), alignment errors were low and ranged between 1.5-3.7 cm.

The two most prominent approaches to identify surface changes in successive point clouds include the identification of 185 homologous objects to calculate displacement fields (Teza et al., 2007; Monserrat and Crosetto, 2008) and direct distance calculation (Rosser et al., 2005). Here, the latter type was applied using the Multiscale Model to Model Cloud Comparison (M3C2) which was specifically designed for orthogonal distance measurement in complex terrain (Lague et al., 2013). M3C2 is frequently used to compute distances between multitemporal point clouds and has been applied in numerous studies

investigating geomorphic change (e.g. Barnhart & Crosby, 2013; Cook, 2017; Esposito et al., 2017; James et al., 2017; Williams et al., 2018). Full details can be found in Lague et al. (2013). Briefly, for comparing two successive point clouds A and B, the M3C2 calculates: (i) a normal vector for any given point $i$ of cloud A by fitting a plane to all neighbouring points $NN_i$ that are within a radius $D/2$ of $i$; (ii) a bounding cylinder of radius $d/2$ with the axis centred at $i$ and oriented normally. Each bounding cylinder isolates subsets of clouds A and B that are projected onto the cylinder axis; and (iii) the distribution of distances along the normal, which is used to calculate mean positions of sub-cloud A ($i_1$) and sub-cloud B ($i_2$). The distance measured ($L_{M3C2}$) between $i_1$ and $i_2$ along the normal direction is stored as an attribute of $i$. The standard deviation of the point distribution within the bounding cylinder (a measure of local roughness) is quantified and combined with the alignment uncertainty to estimate errors and provide a parametric local confidence interval (or level of detection) for each distance measurement. The confidence interval thus represents the sum of different error terms factoring in the cumulative effects of instrumental uncertainty, surface roughness, and alignment uncertainty between point clouds (Hodge, 2010; Soudarissanane et al., 2011). Surface change is considered statistically significant when $L_{M3C2}$ exceeds the local error (confidence interval) and is rejected when $L_{M3C2}$ is smaller than the local error.

Here, a normal scale ($D$) of 5 m was adopted and a projection scale ($d$) of 1.5 m. Plausibility of M3C2 calculations was tested by manually comparing each delineated area of significant surface change (rockfall source area) to computations of the Euclidean nearest-neighbour distance (direct cloud-to-cloud calculation). To calculate rockfall volumes, the plausibility-checked results were reanalysed using the M3C2, and (i) a fixed normal scale ($D$) (orthogonal to the average local terrain surface) to avoid an overlap of the bounding cylinders and thus an overestimation of rockfall volume, and (ii) using a reduced projection scale ($d = 0.25$-$0.50$ m) to avoid integration of unchanged terrain adjacent to the rockfall source area into the distance calculation. Local grids (cell size 5 x 5 cm) containing the $L_{M3C2}$ values of the reanalyses were then created for each rockfall source area and the rockfall volume was computed by grid cell aggregation. The distance measurement errors ($L_{M3C2}$ confidence interval) of the grid cells were aggregated for each source area to estimate the rockfall volume error at one sigma level.

In addition to rockfall volume, the following parameters were determined for each source area: mean slope aspect and gradient, height above glacier surface as well as maximum depth of rock detachment (determined as the maximum Euclidean nearest-neighbour distance between the pre-event and the post-event point cloud). Source areas were differentiated as bedrock (rockwall) or unconsolidated sediments (intra-rockwall sediment deposits) based on shape, inclination, and image colour values. Data gaps due to occlusion are considered negligible for the multitemporal rockwall analysis as obstructions, like deep gullies or protruding spurs that often hamper such analyses in heterogeneous rockwall topography, are rare and scan positions were fixed throughout (except for the final scan at MKW in 2016) for minimising potential detrimental effects from changing incidence angles. Long return periods between surveys, however, increase the chance of superimposition and coalescence effects, i.e. adjacent or subsequent events are sampled as one failure only (van Veen et al., 2017; Williams et al., 2018). To improve readability 'rockfall source areas' are referred to as 'rockfalls'.

## 3.2 Rockwall temperature monitoring

Bedrock temperature was monitored in two deep and four shallow boreholes. Deep borehole T-BN is located at KN about 40 m above the current glacier surface at 2,985 m a.s.l., and was drilled perpendicular to the ~ 45° terrain surface to a depth of 30 m (Fig. 5). Deep borehole T-BW (25 m) is situated at 2,975 m a.s.l. in a W-facing rock slope (~ 40°) not monitored with
225 terrestrial LiDAR (Fig. 1). Borehole temperature was recorded at eleven (T-BW) and twelve (T-BN) different depths with an accuracy of ± 0.03 °C (Platinum Resistance Temperature Detector L220, Heraeus Sensor Technology).

A vertical transect consisting of three shallow boreholes (0.8 m deep) was established in a NE-facing section at KN in September 2015 to investigate bedrock temperatures inside the Randkluft (Fig. 5A). The three boreholes are situated (i) at the Randkluft aperture (at glacier surface level) (T-RK1, Fig. 5B and 5E), (ii) 7 m below the 2015 glacier surface (T-RK2, Fig.
5C), and (iii) 14 m below the 2015 glacier surface (T-RK3, Fig. 5D). Another shallow borehole (0.8 m deep) is located around 5 m above the glacier surface in a ESE-facing section at MKE (T-MKE, Fig. 5A). Temperature in all shallow boreholes is measured with wireless miniature data loggers with an accuracy of ± 0.1 °C (Geoprecision M-Log5W-Rock).

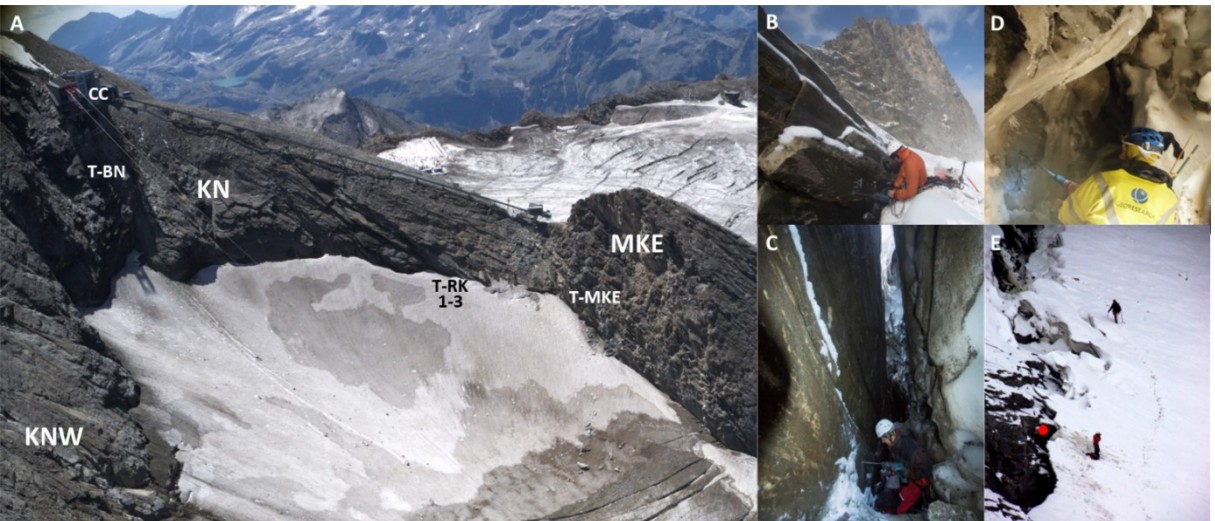

**Figure 5: Deep and shallow borehole temperature monitoring at Kitzsteinhorn. 5A: Air photo (photo: R. Delleske, 24.08.2017); 5B: Measurement site at the Randkluft aperture (T-RK1) (photo: I. Hartmeyer: 04.09.2015); 5C: Measurement site inside Randkluft, 7 m below the 2015 glacier surface (T-RK2) (photo: R. Delleske, 04.09.2015); 5D: Measurement site inside Randkluft, 14 m below glacier surface (T-RK3) (photo: M. Dörfler, 21.09.2018); 5E: View of the investigated Randkluft from the cable car top station (CC); red dot indicates position of T-RK1 (photo: F. Miesen, 04.09.2015).**

## 4 Results

### 4.1 LiDAR data resolution

Data resolution (point density) plays a key role for defining smallest distinguishable detail in point clouds (Hodge, 2010). As a result, more low-magnitude rockfalls will be detected in high-resolution scans compared to low-resolution scans, which introduces issues when rockfall numbers based on scans of differing data resolution are to be compared. To constrain the

influence of data resolution, the mean resulting resolution is compared to the normalised number of rockfalls detected (i.e. the number of rockfalls per 10,000 m² per year) (Fig. 6). A weak positive correlation ($R^2 = 0.18$) can be observed and for rockfalls larger than 0.1 m³ the number of rockfalls is independent of resolution. All further analyses were limited to rockfall volumes above this volume threshold. Compared to other studies, the minimum usable volume of 0.1 m³ derived here is higher than values specified in LiDAR-based change detection surveys using shorter object distances and higher point densities (e.g. Rosser et al., 2007; Williams et al., 2018) but is in good agreement with similar monitoring campaigns carried out in high-alpine settings (e.g. Strunden et al., 2015).

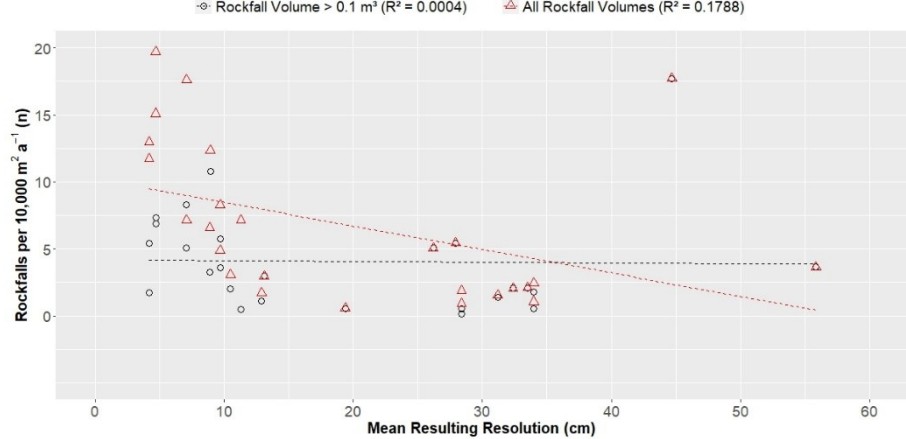

**Figure 6: Detected number of rockfalls per 10,000 m² a⁻¹ plotted against the mean resulting resolution of the performed laserscans. Varying resolutions between acquired scans do not bias the detection of rockfalls larger than 0.1 m³. Dashed lines represent the regression lines of both distributions.**

## 4.2 Rockfall inventory

During the six-year monitoring period (2011-2017) 632 rockfalls were registered with a total volume of 2,564.3 ± 141.9 m³. When omitting rockfalls below the chosen threshold of 0.1 m³ (Sect. 4.1), the total number drops to 374, while the overall volume is reduced only marginally to 2,551.4 ± 136.8 m³ (Table 1). The mean relative error associated with the rockfall volumes is 5.5 % and similar to other high-alpine LiDAR studies which also found single-digit relative errors (Kenner et al., 2011; Strunden et al., 2015). Relative errors are smaller for large rockfall volumes than for small volumes. Uncertainty for rockfalls smaller than 1 m³ is 29.2 %, while for large rockfalls over 100 m³ relative errors drop to 2.2 % due to reduced cumulative effects of instrumental, surface roughness and alignment errors on larger geometries (Hodge, 2010) (see Table S4). Large rockfalls over 100 m³ are rare (n = 5) but account for more than two thirds (68.5 %) of the total volume. The largest registered rockfall has a volume of 879.4 ± 6.3 m³, the volumes of the three next largest rockfalls range between 200-300 m³. With increasing volume an exponential decrease in number of rockfalls can be observed. Small rockfalls below 1 m³ represent 80 % of the total number but account for only 3.7 % of the overall rockfall volume (see companion study by Hartmeyer et al. (2020) for detailed discussion of magnitude-frequency distributions).

**Table 1: Absolute and normalised rockfall numbers (n) and volumes (m³) (> 0.1 m³). Normalised rockfall number (volume) refers to rockfall number (volume) per 10,000 m² per year.**

| | | | TOTAL | KN | KNW | MKE | MKW | MGE |
|---|---|---|---|---|---|---|---|---|
| 0.1 - 1 m³ | Number (n) | Abs. | 299 | 83 | 150 | 15 | 8 | 43 |
| | | Norm. | 2.15 | 5.89 | 1.87 | 2.25 | 3.20 | 1.19 |
| | Volume (m³) | Abs. | 94.7 | 24.5 | 45.9 | 6.7 | 2.8 | 14.8 |
| | | Norm. | 0.7 | 1.7 | 0.6 | 1.0 | 1.1 | 0.4 |
| 1 - 10 m³ | Number (n) | Abs. | 50 | 13 | 21 | 4 | 5 | 7 |
| | | Norm. | 0.37 | 0.92 | 0.26 | 0.60 | 2.00 | 0.19 |
| | Volume (m³) | Abs. | 151.8 | 48.3 | 65.9 | 12.6 | 10.7 | 14.4 |
| | | Norm. | 1.1 | 3.4 | 0.8 | 1.9 | 4.3 | 0.4 |
| 10 - 100 m³ | Number (n) | Abs. | 20 | 5 | 7 | 3 | 2 | 3 |
| | | Norm. | 0.15 | 0.35 | 0.09 | 0.45 | 0.80 | 0.08 |
| | Volume (m³) | Abs. | 547.8 | 104.6 | 156.7 | 54.0 | 136.2 | 96.4 |
| | | Norm. | 4.4 | 7.4 | 2.0 | 8.1 | 54.4 | 2.7 |
| 100 - 1,000 m³ | Number (n) | Abs. | 5 | 3 | 1 | 1 | - | - |
| | | Norm. | 0.04 | 0.21 | 0.01 | 0.15 | - | - |
| | Volume (m³) | Abs. | 1,757.0 | 1,278.0 | 272.7 | 206.3 | - | - |
| | | Norm. | 12.5 | 90.7 | 3.4 | 30.9 | - | - |
| All Rockfalls | Number (n) | Abs. | 374 | 104 | 179 | 23 | 15 | 53 |
| | | Norm. | 2.71 | 7.38 | 2.24 | 3.45 | 6.00 | 1.46 |
| | Volume (m³) | Abs. | 2,551.4 | 1,455.4 | 541.2 | 279.6 | 149.7 | 125.5 |
| | | Norm. | 18.7 | 103.3 | 6.8 | 41.9 | 59.9 | 3.5 |

Frontal views of the monitored rockwalls with indicated rockfall source areas are provided in Fig. 7 and in the supplement (Fig. S1) and show a concentration of rockfalls along heavily fractured, structural weaknesses. Highest rockfall activity was observed at KN, which has developed parallel to the cleavage dip (~ 45° NNE) and is dissected by a highly fractured, slope parallel escarpment running from the cable car summit station to the current glacier margin (Fig. 3B). No rockfall activity was registered in the upper half of the escarpment, whereas in the lower, glacier-proximal half high activity was observed during

the entire monitoring period (Fig. 7A). Three out of the five largest rockfalls in the entire study area detached from this area forming large cubic to rhomboidal blocks of up to 5 m length. The largest of the events occurred on August 18th, 2012 at around 15:00 and was visually and acoustically observed by the cable car staff and by tourists. Immediately adjacent to the glacier several joint-bordered rock bodies were detached resulting in a blockslide (Fig. 8A and B). After detachment, blocks were either retained by the glacier immediately below the source area, or slid over the glacier surface for more than 200 m

carving distinct chutes into the firn covering the glacier (Fig. 8G). The lowest part of the detached rock fragment was covered by snow/firn (Fig. 8A) and was not imaged on the laserscan predating the event (August 2011). The calculated rockfall volume (879.4 ± 6.3 m³) only refers to relief visible above the glacier surface and thus represents a slight underestimate of the true volume.

Rockfall activity at the steep MKE (mean slope 71°) is largely restricted to a zone of highly fractured micaschists that runs

diagonally through the rockwall following the cleavage direction (Fig. 3C). Instability within this weakness zone is highest in

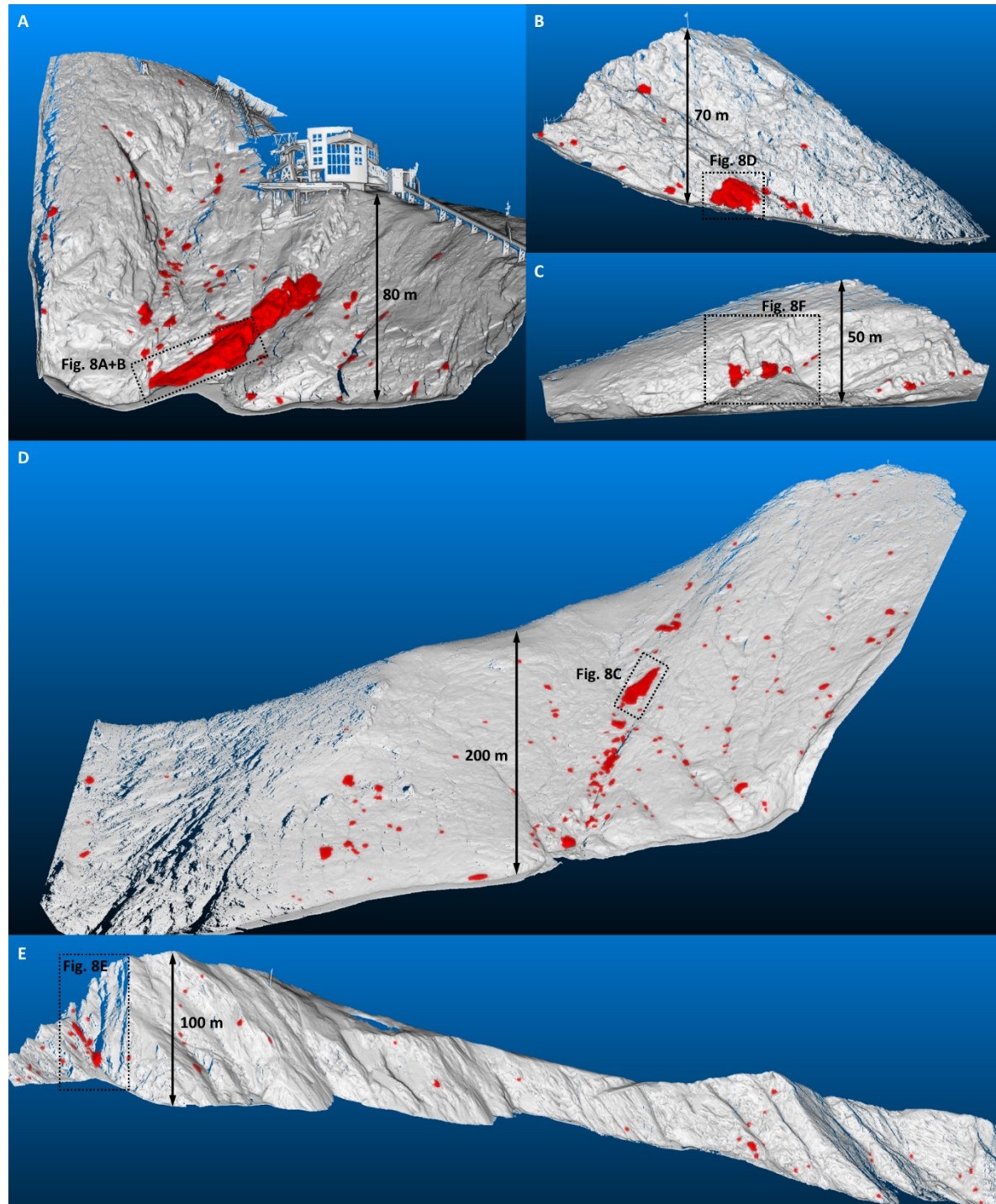

**Figure 7: Frontal views of monitored rockwalls. Rockfall source areas identified during the six-year monitoring period are indicated in red. 7A: Kitzsteinhorn north-face (KN); 7B: Magnetkoepfl east-face (MKE); 7C: Magnetkoepfl west-face (MKW); 7D: Kitzsteinhorn northwest-face (KNW); 7E: Maurergrat east-face (MGE). Displayed topography and glacier level were taken from the most recent LiDAR survey (i.e. from 2016 for MKW and from 2017 for all other rockwalls).**

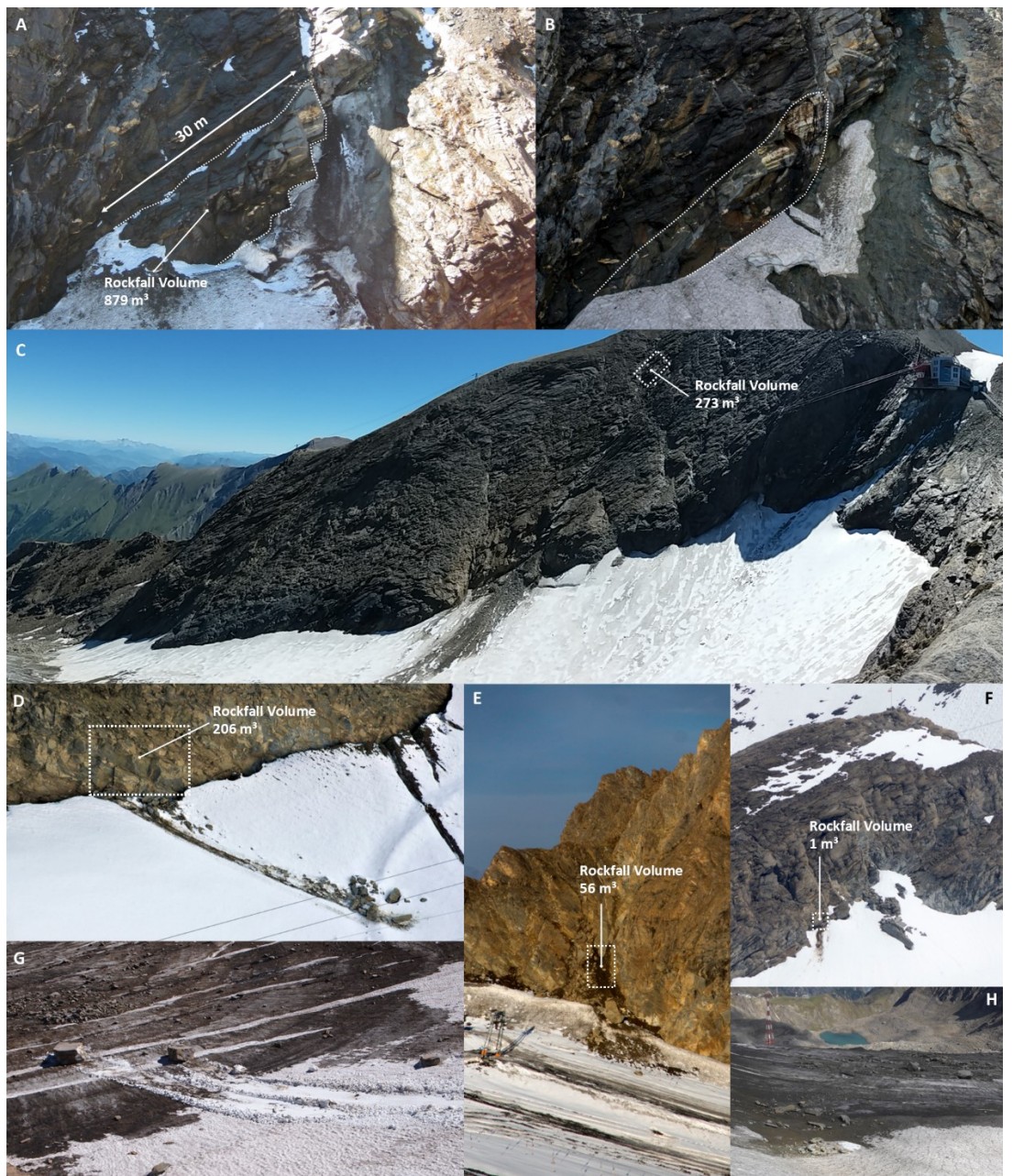

**Figure 8: Documentation of rockfall events for all monitored rockwalls. 8A: Pre-event topography of the largest rockfall registered in the study period at KN (879.4 ± 6.3 m³), the dotted line marks the (still-attached) blocks which were released during the event. The remnant of an ice-apron is visible directly right of the source area (photo: M. Keuschnig, 17.08.2011); 8B: Post-event topography, source area is marked by dotted line. Note the difference in colour between the fresh source area (bright surface) and the terrain above (dark surface) (photo: R. Delleske, 28.08.2019); 8C: Fresh deposits on the glacier surface after 272.7 ± 11.4 m³ (second largest event recorded) of rock detached 97 m above the glacier at KNW (photo: I. Hartmeyer, 26.08.2016); 8D: 206.3 ± 11.6 m³ rockfall at the foot of MKE (photo: I. Hartmeyer, 05.11.2011); 8E: Fresh rockfall deposits below a couloir at south end of MGE after a 56.2 ± 4.1 m³ rockfall (photo: R. Delleske, 12.09.2012); 8F: Small rockfall event (1.3 ± 0.3 m³) at MKW (photo: R. Delleske, 12.09.2012); 8G: Cubic blocks deposited on glacier after event described in 8A (photo: I. Hartmeyer, 17.08.2012); 8H: Extensive cover of old rockfall deposits on the central part of the Schmiedingerkees (photo: I. Hartmeyer, 18.09.2018).**

the immediate vicinity of the glacier surface where on August 27[th], 2011 around 23:00 a 206.3 ± 11.6 m³ rockfall event occurred (Fig. 7B and 8D). The event was registered acoustically by the cable car staff and represents the fourth largest rockfall recorded during the six-year study period. The lower edge of the source area was at glacier surface level, detached blocks were either deposited immediately at the glacier margin or transported towards the glacier's centre for a maximum of 80 m (Fig. 8D). Due to the steepness of the dominant joint sets (J1, J2) the rockfall probably occurred as toppling failure.

At the other side of the Magnetkoepfl (MKW) a steep scarp was created in the centre of the rockwall by several rockfalls that occurred in the decades prior to the start of the measurements (personal communication by cable car staff) (Fig. 7C and 8F). The rockfall scarp remained a prominent source area for rockfall during the monitoring period, including two events > 50 m³ (79.3 ± 7.5 m³, 57.0 ± 4.1 m³) that detached immediately above the talus cone created by past rockfall activity.

At KNW rockfall activity mainly concentrated along a prominent fault across the entire rockwall (Fig. 7D, 8C and 3E), which included the second largest rockfall (272.7 ± 11.4 m³) observed during the monitoring period. Its rockfall source area is located 97 m above the glacier surface and may coincide with the LGM trimline. The exact date of the rockfall is unknown, photos taken on August 26[th], 2016 demonstrate fresh deposits at the glacier surface (Fig. 8C), pointing to an event date in the preceding days or weeks.

Among the rockwalls investigated rockfall activity was lowest at MGE (Table 1). High activity was restricted to an incised couloir at the south end of the rockwall where a 56.2 ± 4.1 m³ rockfall occurred between August 21[st] and September 11[th], 2012. The source area was located 10 m above the glacier surface, detached blocks slid onto the glacier surface and were deposited several tens of metres from the rockwall (Fig. 7E and 8E).

## 4.3 Other mass movements

In addition to rockfalls, 113 source areas were identified in unconsolidated sediments. The total volume of these mass movements is 292.0 ± 72.3 m³. Nine mass movements larger than 10 m³ were identified and account for 56.2 % of the total volume. The size distribution follows the pattern of rockfall volume distribution where smaller mass movements are frequent but represented only a small part of the overall volume and show an exponential decrease in number with increasing volume (Table S5). These types of mass movement were almost fully limited to KNW and KN. The two rockwalls are the least steep and permit accumulation of thin sediment veneers on intra-rockwall couloirs and ledges. Together, KNW and KN display 90.3 % of the total number and 99.3 % of the total volume of all loose sediment movements.

Further mass losses identified relate to ice-apron degradation in a well-shaded location at the lower part of KN adjacent to the glacier surface (Fig. 8A and B). The overall ice loss from 2011-2017 was 575.9 ± 73.9 m³. The single biggest recorded ice loss was 424.1 ± 60.7 m³ between August 2012 and August 2015 (no data acquisition in 2013 and 2014 due to persistent snow cover), followed by mass losses of 66.8 ± 4.6 m³ and 51.4 ± 4.8 m³ between August 2016 and August 2017, and a mass loss of 33.6 ± 3.8 m³ between August 2015 and August 2016. The two mass losses recorded between 2016 and 2017 are underestimated as late snow cover during the second scan obscured the rockwall.

## 4.4 Directional rockfall distribution

Directional analysis of rockwall surface area demonstrates highest shares in the NW-sector (45 %) due to the dominant influence of KNW, followed by E- and N-facing areas (17 % each). S-oriented source areas are largely missing due to the absence of significant S-faces in the study area (Fig. 9A). The bulk of the registered rockfalls originated in N- and NW-oriented rockwall sections between 2,900 and 3,100 m a.s.l.. SE-facing rockfall source areas between 2,800 and 3,000 m a.s.l. represent another distinct azimuth cluster (Fig. 9B). Normalised rockfall volume (i.e. rockfall volume per 10,000 m² per year) is

significantly elevated in the N-, W- and SE-sector and low for all other slope aspect sectors (Fig. 9C). Normalised rockfall numbers (i.e. rockfall number per 10,000 m² per year) peak in the same sectors as normalised rockfall volume (except for the S-sector, which, however, is represented by just a single rockfall) (Fig. 9D). High (normalised) activity in N-, W- and SE-sectors most likely reflects the dominant discontinuity orientations at the study site which promote the detachment of cubic to rhomboidal detachments in N-facing areas and failures along steep joint systems in W- and SE-facing areas (Fig. 3).

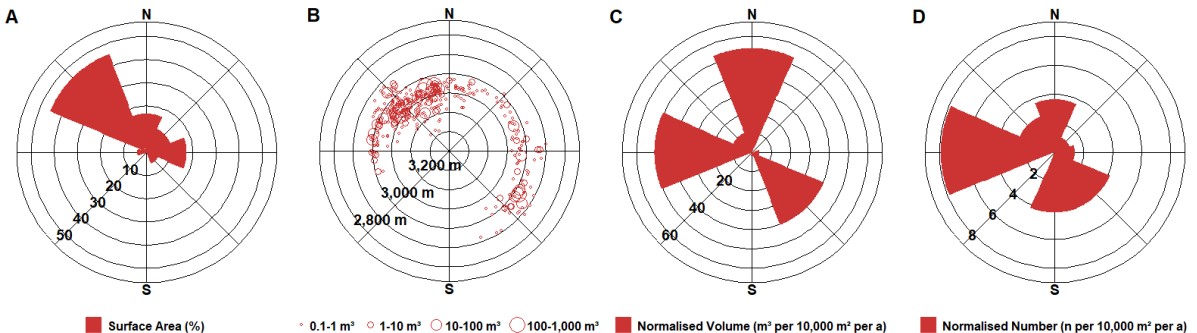


**Figure 9A: Sectoral distribution of surface area. 9B: Slope azimuth angle and elevation in metres above sea level for all rockfalls. 9C: Normalised rockfall volume by sector (m³ 10,000 m⁻² a⁻¹). 9D: Normalised rockfall number by sector (n 10,000 m⁻² a⁻¹).**

## 4.5 Altitudinal rockfall distribution

To further explore rockfall distribution with elevation, we classified the investigated surfaces into vertical 50 m bins. Total

rockwall surface area is almost normally distributed and shows the largest share between 2,900-2,950 m a.s.l. (~ 65,000 m²; 27.0 % of total surface area) (Fig. 10A and Table S6). Normalised and absolute rockfall volumes peak in the same elevation class. Over 37 m³ per 10,000 m² a⁻¹ originated between 2,900-2,950 m a.s.l. (Fig. 10B), which is equivalent to more than half (54.4 %) of the total rockfall volume. The normalised number of rockfalls is highest between 2,950-3,000 m a.s.l. (4.6 rockfalls per 10,000 m² a⁻¹) and declines significantly with increasing/decreasing elevation (Fig. 10C). Absolute rockfall numbers peak

between 2,900-2,950 m a.s.l. and 2,950-3,000 m a.s.l., where approximately two thirds (63.8 %) of the detected rockfalls originate (Table S6).

To detail the vertical distribution of rockfall source areas, the height differences between rockfall source areas and the local glacier surface are calculated and grouped into 10 m bins (Fig. 11A and Table S7). Immediately above the glacier surface (0-10 m) rockfall volumes are by far the highest (75.6 m³ per 10,000 m² a⁻¹) (Fig. 11B). Exactly 60 % of the total rockfall volume

detached from this segment, which constitutes only 15 % of the total rockwall surface area. Another 15 % of the rockfall

volume detached from the next higher segment (10-20 m). With increasing distance from the glacier surface, a sharp decrease in rockfall volume can be observed. In the 10-20 m segment, normalised rockfall volume slightly exceeds 20 m³ per 10,000 m² a⁻¹, while in all other height classes rates remain below 10 m³ per 10,000 m² a⁻¹. Only in two segments (90-100 m, 170-180 m), this pattern is masked by the presence of comparatively large, singular rockfalls.

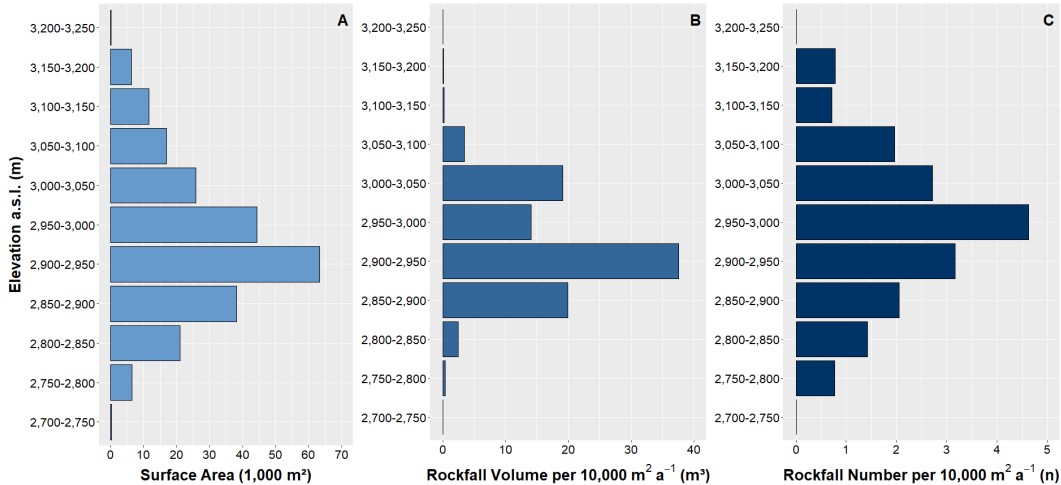


**Figure 10A: Rockwall surface area; 10B: Normalised rockfall volume; 10C: Normalised number of rockfalls; all grouped by elevation in metres above sea level. A distinct peak in rockfall volume is observed between 2,900 and 2,950 m a.s.l..**

Analysed individually, a positive correlation between rockfall volume and proximity to glacier surface occurs for all rockwalls except KNW. The vast majority of the rockfall volume is detected within 10 m of the glacier surface at MGE (73 %), KN

(79 %) and MKE (98 %). Considering the first 20 m above glacier surface the volume percentages exceed 90 % for all three rockwalls.

At MKW rockfall volumes are small in the lowest segment (3 %) and 96 % of the total rockfall volume occurs in the segment above (10-20 m). Here, rockfall activity in decades prior to the start of the monitoring created a steep scarp around 15 m above the current glacier surface. The rockfall deposits, likely several thousand cubic meters of rock, accumulated at the foot of

MKW and constituted a talus cone that decoupled parts of the rockwall from the glacier. Numerous rockfalls detached from this scarp during the study period, indicating continued stress release after the preceding events.

As mentioned, no pronounced glacial proximity pattern was found for KNW, where only 12 % of the rockfall volume detached within the first 10 m. Here, a significant 272.7 ± 11.4 m³ rockfall occurred in summer 2016 (Sect. 4.2) which constituted around half of the total rockfall volume at this site. Still, after excluding this event, only a rather weak proximity pattern is

observed (23 % of the volume within the lowest 10 m) clearly deviating from the patterns observed at the other four rockwalls. Analysis of rockfall numbers confirms the glacial proximity pattern even though the correlation is much less pronounced than for the elevation volume distribution. Highest normalised rockfall numbers (3.9 rockfalls per 10,000 m² a⁻¹) are once again found in the lowest segment (0-10 m) (Fig. 11C). The mean value for all higher segments (i.e. 10-260 m) equals 2.5 rockfalls

per 10,000 m² a⁻¹ with significant variations between the different height classes. Overall 21 % of all rockfalls (78 of 374) occurred in the first 10 m above the glacier surface – a distinct contrast to the dominance of rockfall volumes in that segment. Comparing rockfall numbers across the rockwalls yields diverse results: At KN particularly high rockfall numbers are found between 30 and 50 m above the glacier. KNW shows a more uniform pattern with a rather balanced distribution over the first 100 m and a slight decrease at higher elevations. At MKE, rockfall is restricted to the immediate adjacency to the glacier and above the 0-10 m segment only minimal rockfall activity is observed. At MKW and MGE, most rockfalls occurred within 20 m of the glacier surface (~ 70 % and 90 %, respectively).

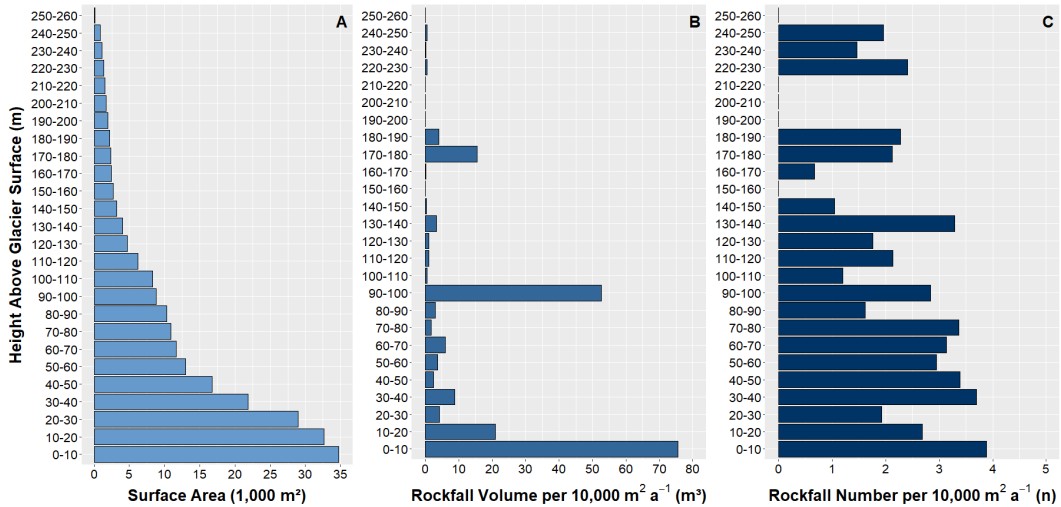

**Figure 11A: Rockwall surface area; 11B: Normalised rockfall volume; 11C: Normalised number of rockfalls; all classified by height above glacier surface. Areas exposed by recent glacier retreat are heavily susceptible to rockfall. During the observation period (2011-2017) 60 % of the total rockfall volume detached within 10 m of the current glacier surface, 75 % detached within 20 m.**

The observed distribution of rockfall magnitudes and frequencies is described by a distinct negative power function over four orders of magnitude (Fig. 12). To test the statistical robustness of the discovered differences between glacier-proximal (< 10 m above glacier) and glacier-distal (> 10 m above glacier) rockfall activity the goodness of fit was analysed using a bootstrapping approach (full details are given in Hartmeyer et al., 2020). In the analysis 20 % of the rockfalls were randomly removed and the data set was resampled 100,000 times to assess the sensitivity of the power-law-exponent $b$, which represents a frequently used variable to characterise spatiotemporal rockfall variation (e.g. Dussauge-Peisser et al., 2002; Barlow et al., 2012). Results demonstrate robust power law exponent $b$ estimates of $0.51^{+0.07}_{-0.05}$ for the proximal and $0.69^{+0.04}_{-0.03}$ for the distal datasets at 95 % confidence level.

To selectively examine the statistical sensitivity to individual rare events, the power-law-fits were recalculated after omitting the five largest rockfalls (volumes > 100 m³). Power law exponents for proximal rockfalls ($0.59^{+0.07}_{-0.05}$) and distal rockfalls ($0.71^{+0.04}_{-0.03}$) only slightly increase and show the significance of the differences observed between proximal and distal areas. Normalised rockfall volume in proximal areas (11.7 m³ per 10,000 m² a⁻¹) was 2.6 times higher than in distal areas (5.2 m³ per 10,000 m² a⁻¹) in the reduced data set (volumes > 100 m³ omitted).

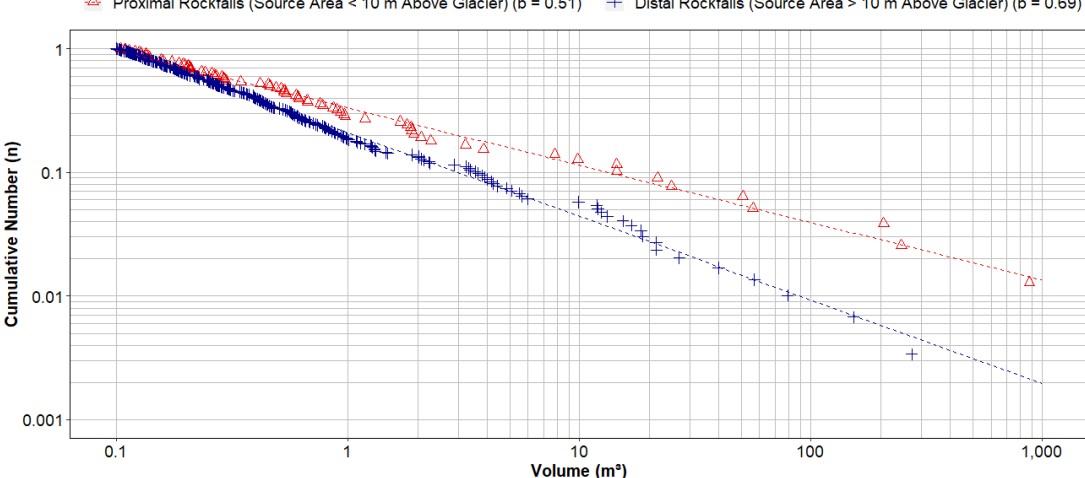

**Figure 12: Magnitude-frequency distributions for glacier-proximal rockfalls and glacier-distal rockfalls. Proximal rockfalls (b = 0.51) are fitted by a significantly flatter regression line than distal rockfalls (b = 0.69) indicating an increased occurrence of large events in recently deglaciated areas.**

### 4.6 Rockfall failure depths

Among the 374 rockfalls identified, depth of failure ranges between 0.17 and 6.45 m. Near-surface failures dominate as 69 % of all rockfalls failed within the top 0.5 m and another 22 % in depths between 0.5 and 1 m. Eleven rockfalls with failure depths of more than 2.0 m were recorded (2.9 %) and only five rockfalls failed in depths larger than 3.0 m (1.3 %) (Fig. 13, Table S8). Classification of rockfall failure depth by slope aspect demonstrates an increased occurrence of relatively deep failures (> 1 m, > 2 m) in W- and SE-facing areas (Table 2). This pattern is consistent with dominant local discontinuities which predispose N-facing rockwall sections to thick dip-slope failures along the NNE-dipping cleavage, as well as W- and SE-facing sections to large (toppling) failures along the steep joint sets J1 and J2 (Fig. 3).

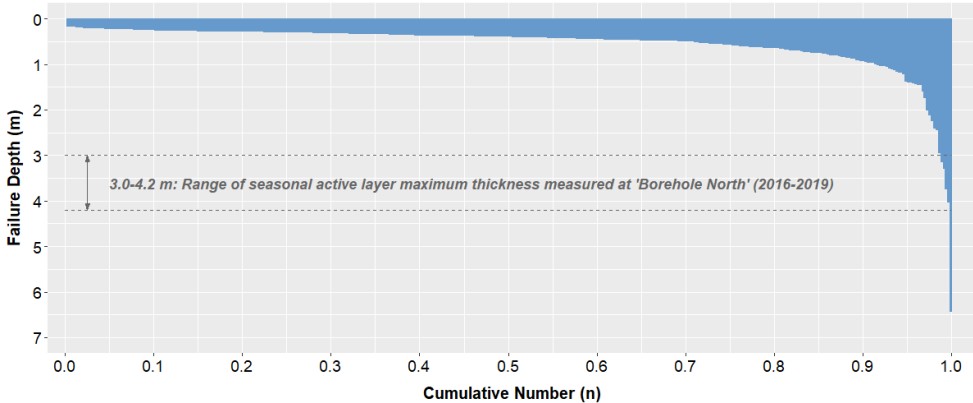

**Figure 13: Failure depths for all registered rockfalls (n = 374). More than 90 % of all rockfalls failed within less than 1 m from the surface. The seasonal active layer maximum thickness measured at T-BN (30 m deep borehole at KN) ranged from 3.0 to 4.2 m (2016-2019).**

**Table 2: Rockfalls with failure depths > 1 m and > 2 m classified by slope aspect. Rockfalls with failure depths > 1 m (> 2 m) are most frequent in W- and SE-facing rockwall sections.**

| Slope Aspect | Failure Depth > 1 m | | Failure Depth > 2 m | |
|---|---|---|---|---|
| | *Total Number (n)* | *Normalised Number (n per 10,000 m² per a)* | *Total Number (n)* | *Normalised Number (n per 10,000 m² per a)* |
| N | 8 | 0.34 | 2 | 0.09 |
| NE | - | - | - | - |
| E | 2 | 0.08 | 1 | 0.04 |
| SE | 8 | 1.07 | 3 | 0.40 |
| S | - | - | - | - |
| SW | - | - | - | - |
| W | 5 | 1.24 | 3 | 0.71 |
| NW | 10 | 0.16 | 2 | 0.03 |

## 4.7 Bedrock temperature

Temperatures recorded at the two deep boreholes (T-BN, T-BW) clearly indicate permafrost conditions. From 2016-19 the seasonal maximum active layer thickness at T-BN varied between 3.0 and 4.2 m (Fig. 14). Active layer formation usually starts in late May, maximum thickness is reached in early September, and complete freezing occurs in early or mid-October. Repeated lightning strike damage resulted in fragmentary data recording at T-BW and hinders full seasonal characterisation of active layer evolution. Seasonal temperature variations at T-BN occur down to a depth of 15-20 m, below which a constant temperature of -1.8 °C was observed over the entire period. At the Kitzsteinhorn W-face (T-BW) temperature at 25 m borehole depth ranged between -1.1 to -1.2 °C (Fig. S2) and comparable values are expected to occur also at E-oriented rockwalls in the study area, given similar topo-climatic conditions at W- and E-faces (Schrott et al., 2012).

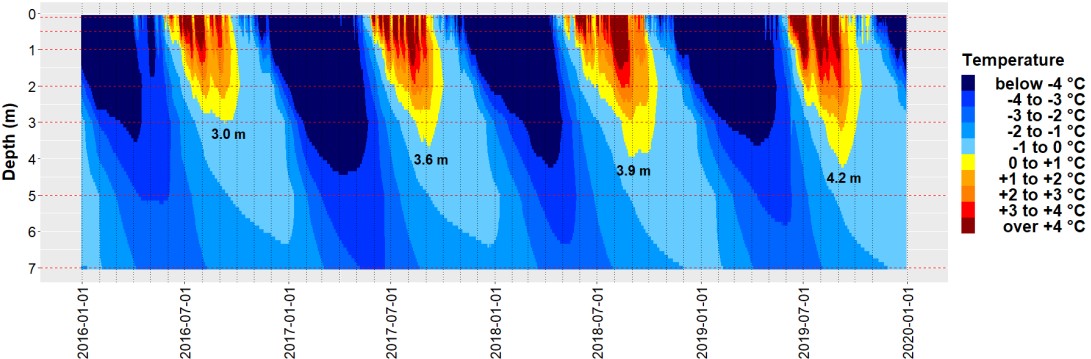

**Figure 14: Four-year borehole temperature record from 30 m deep borehole T-BN (2,985 m a.s.l.) located at Kitzsteinhorn north-face (KN) (first 7 m are displayed). The seasonal maximum thickness of the active layer (indicated on plot) ranged from 3.0 to 4.2 m.**

Shallow bedrock temperatures measured at 0.8 m borehole depth (1.0 m at T-BN) demonstrate significant contrasts between the Randkluft (T-RK2, T-RK3) and the open rockwall (T-BN, T-MKE) (Fig. 15), and indicate a pronounced modification of the ground thermal regime after deglaciation. Temperatures inside the Randkluft remained slightly below or at 0 °C during the entire observation period and show near-isothermal behaviour with annual variations of just around 1 K (Table 3). Significant, short-term autumn cooling through advection of cold air into the open Randkluft was registered only once (Oct 2016) and

during the winter season a slow, long-term cooling trend was observed, which ends abruptly in late spring (May/Jun) most
likely through extensive percolation of meltwater into the Randkluft. Large seasonal amplitudes around 20 K were recorded
in the open rockwall where temperatures ranged from -12 to +6 °C at the N-facing T-BN, and from -12 to +12 °C at the warmer
ESE-facing T-MKE. Seasonal amplitudes at the Randkluft aperture (T-RK1) varied between 7 and 9 K over the four-year
monitoring period.

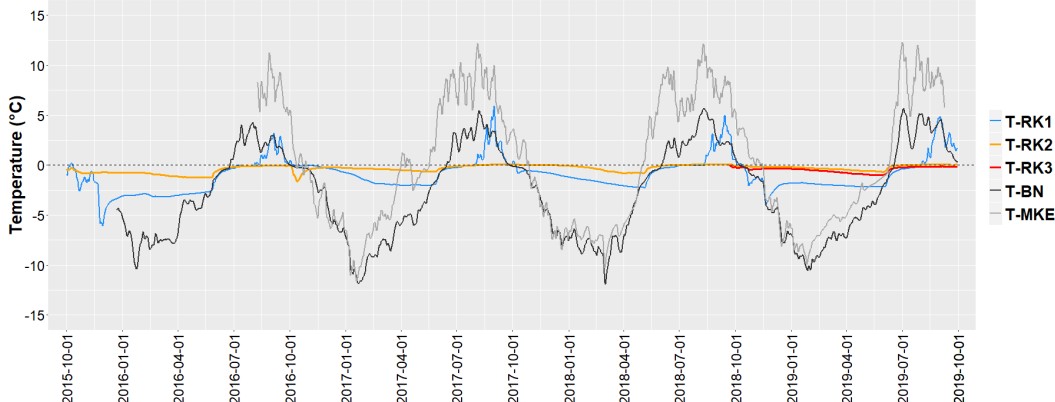

**Figure 15: Near-surface bedrock temperature for five measurement sites located at Kitzsteinhorn north-face (KN) and Magnetkoepfl east-face (MKE). Perennially-frozen, near-isothermal conditions observed inside the Randkluft (T-RK2, T-RK3) contrast with large seasonal amplitudes in the open rockwall. All temperatures were measured at 0.8 m borehole depth (1.0 m at T-BN).**

**Table 3: Annual mean, minimum, and maximum near-surface bedrock temperature for five measurement sites at Kitzsteinhorn north-face (KN) and Magnetkoepfl east-face (MKE).**

| Site | Elev. | Aspect | 2015/2016 1 Oct – 30 Sep | | | | 2016/2017 1 Oct – 30 Sep | | | | 2017/2018 1 Oct – 30 Sep | | | | 2018/2019 1 Oct – 30 Sep | | | |
|---|---|---|---|---|---|---|---|---|---|---|---|---|---|---|---|---|---|---|
| | *m a.s.l.* | | Mean °C | Min. °C | Max. °C | Diff. °C | Mean °C | Min. °C | Max. °C | Diff. °C | Mean °C | Min. °C | Max. °C | Diff. °C | Mean °C | Min. °C | Max. °C | Diff. °C |
| T-RK1 | 2,906 | NE | -1.7 | -6.1 | 3.2 | 9.3 | -0.5 | -2.1 | 5.9 | 8.0 | -0.6 | -2.3 | 5.0 | 7.3 | -1.0 | -3.9 | 4.9 | 8.8 |
| T-RK2 | 2,899 | NE | -0.6 | -1.3 | 0.0 | 1.3 | -0.3 | -1.6 | 0.1 | 1.7 | -0.2 | -0.8 | 0.1 | 0.9 | -0.2 | -0.7 | 0.1 | 0.8 |
| T-RK3 | 2,892 | NE | - | - | - | - | - | - | - | - | - | - | - | - | -0.5 | -1.0 | -0.1 | 0.9 |
| T-BN | 2,985 | N | - | - | - | - | -2.8 | -11.9 | 5.5 | 17.4 | -2.5 | -11.9 | 5.7 | 17.6 | -2.7 | -10.5 | 5.7 | 16.2 |
| T-MKE | 2,902 | ESE | - | - | - | - | 0.2 | -11.6 | 12.2 | 23.8 | 0.1 | -10.3 | 12.1 | 22.4 | -0.6 | -9.9 | 12.3 | 22.2 |

**5 Discussion**

The analysis of long-term terrestrial LiDAR data from two high-alpine cirques shows that rockfall source areas are grouped
along heavily fractured, pre-existing structural weaknesses (Sect. 4.2), in accordance with former studies that found
correlations between fracturing and rockwall retreat (Sass, 2005; Moore et al., 2009). Sectoral analyses of (normalised) rockfall
activity demonstrate increased volumes, numbers, and deeper failure plains in N-, W- and SE-facing slopes (Sect. 4.6), which
is consistent with the orientations of major discontinuities at the study site. Steep joint sets (J1, J2) facilitate large detachments
in W- and SE-oriented terrain, while strike and dip of the micaschist cleavage promote frequent dip-slope failures in N-facing
rockwall sections. Particularly the latter mode of failure may represent a key mechanism of cirque expansion as pronounced

north-south elongated cirque morphologies at the Kitzsteinhorn indicate effective cleavage-driven headwall sapping over long time-scales (Hartmeyer et al., 2020).

Further analysis reveals considerably increased rockfall activity in the immediate proximity (10-20 vertical meters) of the current glacier surface, which emerged from the ice only very recently. While some of the increase may be related to a slight steepening of rockwall gradients towards the glacier surface (Fig. 2), a number of other processes are likely responsible for the observed glacier-proximal rockfall increase.

## 5.1 Antecedent rockfall preparation inside the Randkluft

Slope debuttressing following deglaciation is frequently considered to cause mass movements, particularly in case of larger slope failures. (e.g. Holm et al., 2004; Allen et al., 2010). At the base of the investigated rockwalls, however, seasonally air- or snow-filled voids between glacier and cirque wall ('Randkluft') prevent permanent physical contact between rock and ice and thus effectively hinder debuttressing. The existence of a Randkluft is not site-specific but rather common at alpine (cirque) glaciers (e.g. Gardner, 1987; Mair and Kuhn, 1994; Sanders et al., 2012). Among the rockwalls investigated here, Randkluft

systems are most pronounced below KN, possibly due to the principal flow direction of the adjacent glacier perpendicularly away from the slope. Randkluft development is rather limited at KNW, likely caused by substantial (avalanche) snow accumulation at the foot of the tall, low-gradient rockwall.

Local Randkluft systems at the Kitzsteinhorn are usually open during late summer/early fall (Fig. 16A), even though Randkluft width and depth exhibit considerable interannual variations. It is evident from our observations that the debuttressing effect, if

relevant at all (McColl, 2012; McColl and Davies, 2012), can occur subglacially only, in the lowermost parts of the Randkluft. Sporadically, the collapse of ice bridges may cause small-scale debuttressing locally, but in general this mode of failure seems not too effective. Debuttressing can also not explain the increased rockfall activity several meters above the glacier surface, i.e. in areas already ice-free for years or decades.

Direct observations from Randkluft environments are scarce and have so far relied on visual evidence (Johnson, 1904) and

some in-situ air temperature records (Battle and Lewis, 1951). Assessments of the thermal regime range from freeze/thaw-dominated conditions (Johnson, 1904) to stable sub-zero conditions (Gardner, 1987). Here, we report a first set of bedrock temperatures from a Randkluft: four-year records from shallow boreholes (0.8 m deep) located 7 m and 14 m below glacier surface (Sect. 4.7). Temperatures remain at or just below 0 °C during the entire observation period and display extremely low seasonal variability (~ 1 K). We visually observed significant melt- and rainwater runoff from the rockwall into the Randkluft

during the summer season. Below Randkluft depths of 5 to 10 m rockwalls were covered in thick layers of refrozen water ('verglas') that persisted during the entire observation period (Fig. S3).

Perennially frozen conditions inside the Randkluft in combination with the extensive water supply from the rockwall above may significantly increase the efficacy of frost weathering in such subglacial cirque wall sections. One of the few quantitative studies indicates particularly effective rock-fracturing driven by ice segregation within the Randkluft of a temperate glacier in

British Columbia, Canada (Sanders et al., 2012). This observation has recently been substantiated by numerous field and lab

experiments demonstrating intense frost cracking by ice segregation at temperatures just below 0 °C (Girard et al., 2013; Duca et al., 2014; Murton et al., 2016) and thermo-cryogenic rock fatigue due to damage accumulation over longer time scales (Jia et al., 2015). Subcritical stress propagation due to ice segregation driven by sustained freezing and sufficient water supply (Jia et al., 2017; Draebing and Krautblatter, 2019), and high quarrying-related tensile stresses caused by refreezing meltwater at

the bottom of the Randkluft (Lewis, 1938; Hooke, 1991) are therefore hypothesised to be the dominant antecedent processes of rockfall preparation. The special weathering conditions may prepare the high fragmentation of near-Randkluft bedrock which efficiently predisposes cirque walls to shallow failures and ultimately controls the high post-glacial rockfall activity.

Observations made in the present study contribute to a more than one century long discussion on the mechanisms of cirque headwall retreat (Richter, 1900; Martonne, 1901). Earlier studies postulate high erosion at the base of the headwall to account

for the development of the characteristic break of slope (schrundline) and a low-gradient cirque floor (Evans, 1997). We found smooth, vertical rockwall sections inside the Randkluft (Fig. 16A) that do not match the cataclinal headwall morphology above the glacier (~ 45° slope following the direction of cleavage), confirming a vastly different erosion regime at the ice-covered headwall base. Furthermore, the observed ample water supply and refreezing in the narrow, lower Randkluft sections may efficiently strengthen the bond between headwall and glacier ice and thus promote erosion by quarrying, leading to localised

erosion at the lower headwall (Hooke, 1991).

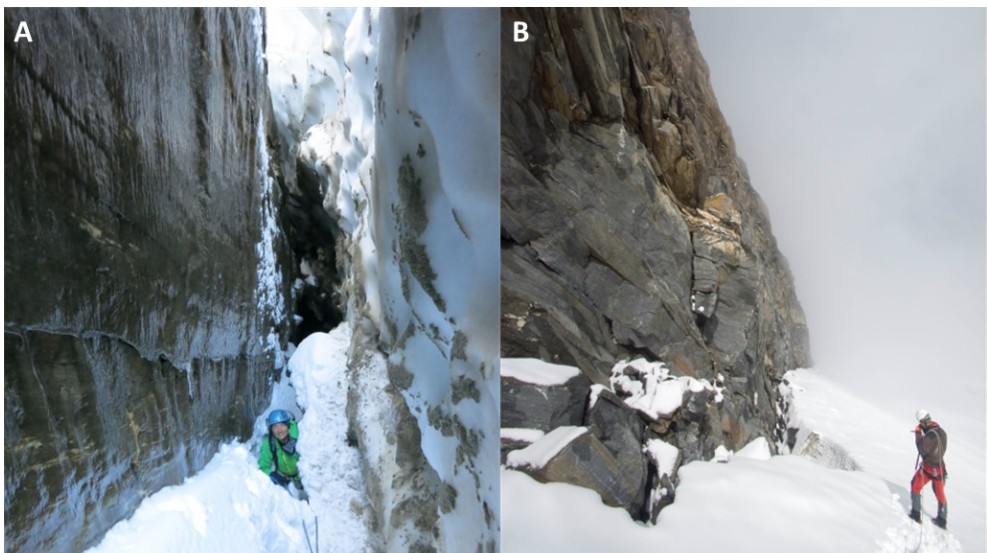

**Figure 16A: Inside the Randkluft below KN (Kitzsteinhorn north-face). Pictured person is standing approximately 8 m below the glacier surface. Continuous ice coating (verglas) on the cirque wall (left half of the photo) indicates permanently frozen conditions inside the Randkluft (Photo: Ingo Hartmeyer, 09.10.2015). 16B: Magnetkoepfl east-face (MKE) and adjacent glacier separated by**
**Randkluft. Recently deglaciated, unstable blocks are visible in the first meters above the glacier surface. Occasionally rockfall deposits are wedged between rockwall and Randkluft lip (bottom left) (Photo: Robert Delleske, 04.09.2015).**

**5.2 Deglaciation-induced thermomechanical forcing and active layer formation**

As glaciers are wasting down, boundary conditions at freshly exposed rockwall sections shift from subglacial to subaerial. The thermal effects of this transition – quantified here for the very first time – are drastic as ground thermal conditions emerge

from a near-isothermal, subglacial setting and convert to a strongly seasonal regime. A modification clearly justifying its recent designation as 'paraglacial thermal shock' (Grämiger et al., 2018). Once ice-free, the measured strong diurnal and seasonal variations are likely to induce pronounced thermal stress leading to deformation (Hasler et al., 2012; Weber et al., 2017) and potentially to failure along critically-stressed discontinuities (Hall, 1999; Gischig et al., 2011). Additionally, cyclic freeze-thaw action will cause rock fatigue (Jia et al., 2015), hydrofracture (Davidson and Nye, 1985; Sass, 2004) and the expansion of water-filled joints (Matsuoka and Murton, 2008), all of which promote destabilization in recently deglaciated rockwall sections (Draebing et al., 2017).

Active layer deepening, a key element of permafrost degradation (Ravanel et al., 2017), significantly alters rock- and ice-mechanical properties (Davies et al., 2001; Krautblatter et al., 2013) and is frequently considered in high-alpine rockfall analyses (e.g. Gruber and Haeberli, 2007; Weber et al., 2019). Failure depth of rockfalls related to permafrost degradation is expected to equal or exceed maximum active layer thickness. At a local borehole monitoring site in a N-facing rockwall section (T-BN) the active layer depth varies between around 3 to 4 m inter-annually. Based on these values, only 0.5 % (below 4 m) to 1.3 % (below 3 m) of all rockfalls failed at greater depths than the maximum thickness of the seasonal active layer. Volume shares are significantly higher due to the large size of the deeper-seated events: Rockfalls with failure depths larger than 3 m (4 m) constitute 44 % (60 %) of the total rockfall volume, suggesting that permafrost degradation could indeed have a substantial impact on total rockfall volume.

Active layer thickness is expected to vary strongly across the investigated rockwalls, mainly due to slope aspect (Schrott et al., 2012), topographic effects (Gruber et al., 2004), and snow cover variations (Haberkorn et al., 2015). Active layer depth monitored at a single borehole (T-BN) is therefore unlikely to be representative for the entire study area. Temperature measurements at T-BW and T-MKE confirm this assumption and point to larger active layer depths at W- and E-facing rockwalls. Particularly for recently deglaciated rockwall sections, permafrost dynamics are poorly understood due to the complex local interplay of glaciological, meteorological and geological controls (Draebing et al., 2014). Bedrock temperatures measured inside the Randkluft (T-RK2, T-RK3) below a NE-facing rockwall are below or at 0 °C and demonstrate the complete absence of an active layer. At the Randkluft aperture (i.e. at the level of the glacier surface) a short 1-2 month time-window with positive temperatures was recorded indicating the formation of a shallow active layer. Glacial downwasting uncovers permanently frozen rockwalls and causes the formation of an incipient active layer, which is likely initiated in the uppermost metres of the Randkluft. This process is expected to alter rock- and ice-mechanical properties (Davies et al., 2001; Krautblatter et al., 2013), promote the infiltration of water (Gruber and Haeberli, 2007; Hasler et al., 2011), and will therefore contribute considerably to the increased rockfall activity near the current glacier surface. Further influences that potentially contribute to high glacier-proximal rockfall activity, include late-spring ground avalanches and channelized rainwater runoff after heavy precipitation. Visual observation suggests strong erosive effects for these processes in the freshly deglaciated sections where blocks at failure stability limit are abundant (Fig. 16B). More precise quantification of such processes would require significantly shorter survey return periods.

# 6 Conclusions

We present a unique rockfall inventory from a six-year terrestrial LiDAR campaign (2011-2017) for permafrost-affected rockwalls of two glaciated cirques in the Central Alps of Austria (Kitzsteinhorn). The five rockwalls studied are all influenced by significant glacial downwasting and ice-apron degradation. We draw the following conclusions:

- The inventory represents the most extensive dataset of high-alpine rockfall to date and the first quantitative documentation of a cirque-wide erosional response of glaciated rockwalls to recent climate warming.
- During the monitoring period 632 rockfalls with an overall volume of $2,564.3 \pm 141.9$ m³ were recorded. In addition, 113 source areas for mass movements with a total volume of $292.0 \pm 72.3$ m³ were detected in unconsolidated sediments. Mass loss from ice-apron degradation accounted for an overall volume of $575.9 \pm 73.9$ m³.
- Rockfall activity concentrates along pre-existing structural weaknesses and was highest in recently deglaciated areas: 60 % of the rockfall volume originated from source areas located fewer than ten meters above the current glacier surface; 75 % detached within 20 meters above the glacier surface.
- Increased mass wasting activity in recently deglaciated areas, such as discovered in the present study, is typical of paraglacial environments, where slope systems gravitationally adjust to new, non-glacial boundary conditions.
- Previous studies on the paraglacial adjustment of bedrock slopes mostly focused on high-magnitude events such as rock avalanches and rockslides, which commonly respond to deglaciation on centennial to millennial timescales. The lower end of the paraglacial magnitude-frequency spectrum is currently poorly characterized. The present study bridges this gap and for the first time provides field evidence of an immediate, low-magnitude paraglacial response in a currently deglaciating rock slope system.
- Distinct Randklufts, which separate the investigated cirque walls from the adjacent glacial ice, effectively prevent debuttressing. To characterise the thermal regime of the Randkluft we carried out unprecedented shallow borehole (0.8 m) measurements at 7 and 14 m Randkluft depth, and found perennially frozen conditions and extensive refreezing of meltwater supplied from the rockwall above.
- Sustained freezing along with sufficient water availability in the Randkluft likely drive subcritical stress propagation by ice segregation and cause high quarrying-related tensile stresses, which contribute to antecedent rockfall preparation when the rockwall is still ice-covered.
- As the glacier is wasting down strong diurnal and seasonal temperature variations induce pronounced thermal stress, cause rock fatigue and lead to the first-time formation of an active layer, which is expected to exert a significant destabilizing effect on glacier-proximal areas.

**Data availability.** The rockfall inventory can be downloaded from the *mediaTUM* data repository under the following weblink: https://mediatum.ub.tum.de/1540134. Terrestrial LiDAR data is available on request.

**Supplement.** The supplement is provided in a separate file.

**Author contributions.** MKE, LS and JO initiated the underlying research project in 2010 and obtained the funding. IH, MKE and RD developed the idea and designed the study. IH and RD conducted the data acquisition and IH analysed the data. All authors contributed to the discussion and interpretation of the data. IH drafted the manuscript with significant contributions from MKR and AL.

**Competing interests.** The authors declare that they have no conflict of interest.

**Acknowledgements**. We would like to thank Alison Anders, Jan Beutel, Robert Kenner, and Associate Editor Arjen Stroeven for their thoughtful feedback and constructive reviews. A. Schober, H. Kugler, M. Dörfler, and F. Miesen kindly provided the photos used for Fig. 3D, 4B, 5D, and 5E.

**Financial support.** This study was co-funded by the Austrian Academy of Sciences (ÖAW) (Project 'GlacierRocks'), the Arbeitsgemeinschaft Alpenländer (ARGE ALP) (Project 'CirqueMonHT') and the Austrian Research Promotion Agency
(FFG) (Project 'MOREXPERT'). We furthermore thank the Gletscherbahnen Kaprun AG (Project 'Open-Air-Lab Kitzsteinhorn') for financial and logistical support.

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
