# Peer review of "Current glacier recession causes significant rockfall increase: The immediate paraglacial response of deglaciating cirque walls"

_Earth Surface Dynamics, 2020_

## Referee Comment (RC1) · Alison Anders (Referee) · 27 Mar 2020

This manuscript presents a fantastic data set with some very interesting results - I believe that it is likely to be widely read and help move the science forward. I do not have expertise to comment on the data collection/processing methodologies, but they were fairly easy to understand as presented and I don't see any big problems. I would encourage the authors to use the observations to more carefully test some hypotheses on what controls the rate of cirque-headwall rockfalls. Specifically, please elaborate on the pattern of rockfall with orientation. The bedrock is described as having prominent cleavage to the NE and lots of jointing (without orientations provided). The majority

of rockfalls are on NW or N facing slopes - which is attributed to specific locations? Perhaps the figure could be clarified to differentiate between rockfalls in different locations. Can this be clarified? There are faults in the vicinity of the major rockfalls? Can those faults be shown on the images? Can joint/fracture spacing be quantified on the different faces? (Perhaps using the exisiting Lidar data?) Is there a difference in the thickness of the rockfalls with orientation (or a pattern related to location)? I would expect that microclimates would impact the rate of development of the active layer and the ultimate thickness of the active layer - and I would also expect that the microclimates might be largely controlled by aspect. Could the diurnal/seasonal temperature variations be modeled for the rockwall faces? Are any instumental temperature data available? Basically, I think that this data set could be probed in more detail to discern the influence of structure (jointing, foliation, faults) vs. climate in the rates and size of rockfalls. Doing so has the potential to increase the impact and significance of the paper.

---

## Author Comment (AC1) · 29 Mar 2020

We thank Alison Anders for her constructive and thoughtful comments. We will respond in more detail after the discussion period and will integrate her suggestions in a revised version of the manuscript.

Several of the points raised are covered in the companion study "Enhanced rockwall retreat and modified rockfall magnitudes/frequencies in deglaciating cirques from a 6-year LiDAR monitoring" (https://www.earth-surf-dynam-discuss.net/esurf-2020-9/), especially the relationship between the discontinuity orientation and the rockfall activity observed. In the revision we will strengthen the link between the two companions. A

far more detailed study on rock temperature, active layer evolution, and microclimate is currently underway.

---

## Short Comment (SC1) · 5 Apr 2020

This work is an impressive study documenting a laborious long-term study of high-alpine mass wasting (due to glaciation) based on repeat terrestrial laser scans as well as an intimate knowledge of the area studied. The manuscript is pleasant to read and informative, however i have a few questions and comments that would substantially enhance the material presented and sharpen the conclusions drawn out of this work. These are in most cases not requiring extra work to be carried out but rather clarifications and possibly additions of the current manuscript. I will try to elaborate on the basic questions in the following as well as specific and detailed comments in a second

part of this commentary.

(i) Specifically I am lacking a pertinent discussion as to the accuracy and validity of your quantitative analysis presented. in no way do i doubt your figures, but in the form they are presented it remains largely unclear how large or small your errors presented are w.r.t. the state of the art and what this errors depend on. I am especially worried since there is no apparent attempt to validate at least part of the figures presented. is it possible to manually cross check the volumes presented with photographs, site visits, deposits on the glacier surface such as seen in fig 4/KNW and KN?

(ii) Exposition, role of the sun/radiation: I am missing a detailed discussion of the exposition and the role of radiation/shading. can you add into figure 5 at which expositions you actually have rock walls in your portfolio and possibly also how much? You show altitude in great detail (fig 6) but little is shown w.r.t. south/north facing. Also your discussion of agrading/degrading permafrost/active layer is weak and in parts not concise w.r.t. the influence of radiation and the stresses originating from it.

(iii) In section 5 you discuss that groups of rockfall can be observed e.g. near structural weaknesses or in immediate proximity of the glacier surface. It would be very interesting to see this observation also in your evidence. can you point out such weaknesses in the topography/photos? can you point out such hot spots in there as well. Fig 4 only shows the approximate distribution and size of the observations. but if you are really able to bin these into classes and connect them with properties of the environment, you should really show evidence for that. maybe only in the form of a spot check and not a total cumulative analysis but without further backing this claim is hard to make. A more detailed discussion and evidence of fracture/weaknesses existing in the kitzsteinhorn rockwalls would be helpful.

(iv) I am not convinced of the discussion of the randkluft as you present it. to my understanding the key property of such deep reaching voids, typically found at the upper boundaries of glacier cirques (not just in the Alps) is that there is no continuous

physical contact between ice and ground (rock). This means that there is no mass loading with ice or water pressure and the rock surface is largely exposed to air. So in effect the rock walls are "free standing" compared to vertical (or steep) rock parts that are completely encased in ice. Due to this missing mass loading and the missing water pressure the hydraulic regime changes (see e.g. Simon Loew et al Aletsch Glacier etc). Due to the Randkluft reaching deep this is probably the case since a long time - a very long time. Concerning the air and the governing temperature regime i disagree that there is no active layer. It may not be very significant but your claim about ice cover in mid october is not convincing, knowing that mid summer is in the end of june and that there is a lot of running water traversing these rock faces from spring (snowmelt) to fall bringing a lot of thermal energy deep into these rock faces below the glacier surface. I rather think that the active layer (and permafrost) regime is of very different properties (temporal, dimensional as well as thermal) as in free surfaces.

So maybe you can add thermal data to back up your evidence. in a minumum this should be MAAT, MAGST etc. a discussion of north/south, shaded vs. unshaded etc.

Minor comments:

Figure 2: Possibly this figure could be augmented by an even newer picture (end of the study period) to show explicitly the deglaciation that took place during the study period. Also, can you quantify this deglaciation somehow?

Page 6/L 149f: The sentence about rope access is irrelevant for this study. leave it out.

Table 2: Rather than repeating the sales brochure of Riegl please specify the settings used for obtaining your data. The general specs of this instruments are known/accessible through the manufacturer to everyone.

Section 3.2: It would help if you give a short synopsis of the algorithms used (M3C2) and not only list benefits similar as how you briefly explain ICP above.

Availability of the data: Is the LIDAR (airborne and terrestrial) available? or can it be

made available.

Page 7, L 183: In addition, THE...

Section 4.1: - What is the detection limit mentioned? And how is this error determined? can you explain what influences this error (besides the size of the rockfall)? - You mention a theoretical discrepancy w.r.t. the detection. can you please detail here? and besides theory, what does it mean in practice for your study?

Figure 3: What are the two lines? The correlations? Please explain this in detail.

Section 4.2: Errors of +/- 1.5 mˆ3 and 1.3mˆ3 respectively. How sure are you? How did you validate this.

Page 12, L 276: You discuss an event that took place before your campaign. If there is a direct context with the observations during your campaign, please explain and back this up with data and plots. If not leave this out. the discussion here is only of a qualitative nature.

Figure 8: Normally CDF functions are given normalized to 100% and not in absolute numbers. Maybe you can also add the thermal data you have to this plot, although clearly one borehole somewhere else is only of limited use in the discussion (you discuss this somewhere later).

Page 14, L 320: I do not see what hinders debuttressing in this case

---

## Author Comment (AC2) · 12 Apr 2020

Dear Jan,

thank you very much indeed for your detailed comments on our manuscript. We will include the suggested improvements in the revision.

---

## Referee Comment (RC2) · Robert Kenner (Referee) · 16 Apr 2020

Dear Authors, Dear Editor,

With great interest, I read the paper of Ingo Hartmeyer et al. This valuable study and the Phanatic dataset provide new insights in a poorly investigated research field. Personally, it was an inspiration for me, as we collected a similar dataset over the last 10 years which is waiting for analysis. I had a couple of comments of which the most are of technical nature, they can be found in the attached PDF of the manuscript. There are 3 larger remarks which probably require some more significant changes but I am sure this will be no large challenge for the authors:

[Figure]

1. You did a good job in estimating the accuracy of your measurements. However, I miss something similar for the statistics. Some of the statistical analysis are based on very small sample sizes or are strongly influenced by single events. I wonder if all this is significant and would appreciate something like a sensitivity analysis.

2. The second part of 5.1 when you describe subglacial rock fracturing is not convincing. First you say, that there is no active layer here (I agree with Jan that there is one at least in the upper meters but likely clearly thinner than above the glacier line) and that the melt water refreezes at the surface but then you write, that large amounts of melt water enables frost cracking. How should that be possible from the surface? Important for subglacial rock weathering within a Bergschrund is probably ice-segregation but this is not even mentioned. This leads me to point 3:

3. Section 5.1 gives the subliminal impression that rock wall erosion or at least rock wall weakening is higher in the Randkluft than on a not glacially affected rock. This is however the major question which is not satisfyingly discussed: If I imagine a constant slope and put a glacier at its foot and than I wait for 50'000 years, the rock wall is commonly steeper in the area around the Bergschrund after this time. These steep belts are called Schrundlines and you made a similar observation at your site. Now where does this steepening come from? Is it because the glacier erodes the foot of the rock wall below the Randkluft or does the glacier in contrast protect the rock wall from temperature forcing and decelerates rock erosion while higher erosion rates above the randkluft cause a flattening of the rock wall? Obviously, erosion rates increase after glaciation right? But what is higher? Subglacial erosion or erosion in summit regions not affected by glaciation? Perhaps you can discuss this very basic but unsolved question.

I am looking forward to a successfull publication!

Robert

Please also note the supplement to this comment:
https://www.earth-surf-dynam-discuss.net/esurf-2020-8/esurf-2020-8-RC2-supplement.pdf
* * *
**ESurfD**

**Supplement:**

**Current glacier recession causes significant rockfall increase: The immediate paraglacial response of deglaciating cirque walls**

Ingo Hartmeyer1, Robert Delleske1, Markus Keuschnig1, Michael Krautblatter2, Andreas Lang3, Lothar Schrott4, Jan-Christoph Otto3

[revised manuscript text omitted]

---

## Author Comment (AC6) · 25 Jun 2020

Dear reviewers, dear editor,

thank you for your highly valuable and thought-provoking comments. We posted point-by-point responses to your comments and uploaded the revised manuscript (one version with markups showing revisions, the other with a clean layout).

Following suggestions by the referees we (i) added new temperature data to the manuscript including a set of unique bedrock temperatures acquired inside the Rand-kluft (Sect. 3.2, 4.7), (ii) expanded the analysis of rockfall with orientation (Sect. 4.4),

and (iii) provided additional information on deglaciation trends during the study period and in the decades prior to the monitoring (Sect 2.1).

Kind regards Ingo Hartmeyer (on behalf of all co-authors)
* * *

---

## Author Response (AR1)

Dear Alison Anders,

5   thank you for your constructive and insightful comments which we feel helped to improve the manuscript.

We (i) uploaded a revised version of the manuscript (one version with markups showing revisions and responses to minor comments, the other with a clean layout), (ii) posted a new author's comment to inform on general amendments in the manuscript, and (iii) below provide a point-by-point response to your comments. Our response is structured as follows: (1)
10   referee comment and (2) author's response including direct references to manuscript changes.

      (1)  I would encourage the authors to use the observations to more carefully test some hypotheses on what controls the rate of cirque-headwall rockfalls. Specifically, please elaborate on the pattern of rockfall with orientation. The bedrock is described as having prominent cleavage to the NE and lots of jointing (without orientations provided).
15           The majority of rockfalls are on NW or N facing slopes - which is attributed to specific locations? Perhaps the figure could be clarified to differentiate between rockfalls in different locations. Can this be clarified?
      (2)  We added information on discontinuities/jointing to the study site description (see Sect. 2 and new Fig. 4), and added a new subchapter in which we elaborate on the pattern of rockfall with orientation (see Sect. 4.4 Directional Rockfall Distribution). We furthermore added a paragraph on the sectoral rockfall distribution to the beginning of
20           the discussion (Sect. 5)

      (1)  There are faults in the vicinity of the major rockfalls? Can those faults be shown on the images?
      (2)  Faults/Weakness zones are now displayed on Fig. 4, 7 and 8 which were added to the manuscript.

25       (1)  Can joint/fracture spacing be quantified on the different faces? (Perhaps using the existing Lidar data?)
      (2)  The quantification of joint/fracture spacing would require higher data resolution. We added new figures (Fig. 4, 7 and 8) to visualise weakness zones in more detail and added a verbal description of zones with increased rockfall activity to Sect 4.2.

30       (1)  Is there a difference in the thickness of the rockfalls with orientation (or a pattern related to location)?
      (2)  We expanded Sect. 4.6 (Rockfall Failure Depths) and added a classification of rockfall failure depth by slope aspect (Table 2).

      (1)  I would expect that microclimates would impact the rate of development of the active layer and the ultimate
35           thickness of the active layer - and I would also expect that the microclimates might be largely controlled by aspect. Could the diurnal/seasonal temperature variations be modeled for the rockwall faces? Are any instrumental temperature data available?
      (2)  We added new (instrumental) temperature data to the manuscript, which includes (i) a set of unprecedented rock temperature data from the Randkluft, and (ii) datasets from 'cold' N-facing sections and 'warm' ESE-facing
40           sections to compare the spectrum of seasonal variations. Data acquisition and measurement sites are described in Sect. 3.2, results are described in Sect. 4.7, and discussed in Sect. 5.1 and 5.2.

      (1)  Basically, I think that this data set could be probed in more detail to discern the influence of structure (jointing, foliation, faults) vs. climate in the rates and size of rockfalls. Doing so has the potential to increase the impact and
45           significance of the paper.
      (2)  By adding the above mentioned information on bedrock temperature and structural weaknesses we hope to give a more rounded picture that allows better differentiation between the influence of structure vs.

**Response to RC2 (Robert Kenner):**

Dear Robert Kenner,

thank you for your constructive and insightful comments which we feel helped to improve the manuscript.

We (i) uploaded a revised version of the manuscript (one version with markups showing revisions and responses to minor comments, the other with a clean layout), (ii) posted a new author's comment to inform on general amendments in the manuscript, and (iii) below provide a point-by-point response to your comments. Our response is structured as follows: (1) referee comment, (2) author's response including direct references to manuscript changes.

(1) You did a good job in estimating the accuracy of your measurements. However, I miss something similar for the statistics. Some of the statistical analysis are based on very small sample sizes or are strongly influenced by single events. I wonder if all this is significant and would appreciate something like a sensitivity analysis.

(2) Following your suggestion we added a sensitivity analysis to the manuscript (see last two paragraphs of Sect. 4.5 and new Fig. 12). The sensitivity analysis is described in full detail in the companion paper.

(1) The second part of 5.1 when you describe subglacial rock fracturing is not convincing. First you say, that there is no active layer here (I agree with Jan that there is one at least in the upper meters but likely clearly thinner than above the glacier line) and that the melt water refreezes at the surface but then you write, that large amounts of melt water enables frost cracking. How should that be possible from the surface? Important for subglacial rock weathering within a Bergschrund is probably ice-segregation but this is not even mentioned.

(2) We added a multi-year temperature dataset from the Randkluft to describe its thermal regime in more detail. Data acquisition and measurement sites are described in Sect. 3.2, results are described in Sect. 4.7, and discussed in Sect. 5.1 and 5.2. Our measurements/observations suggest sustained freezing and ample water supply inside the Randkluft at the same time (during summer) – two key requirements for ice segregation (and thus for frost cracking) (see also analyses of Sanders et al. 2012, Alley et al. 2019, Evans et al. 2020 who all assume enhanced frost cracking inside the Randkluft). We feel that some confusion was arising from our use of the term "subcritical fracture propagation" (which results from ice segregation). We adapted the terminology to emphasise the role of ice segregation more clearly (see amendments in Sect. 5.1).

(1) Section 5.1 gives the subliminal impression that rock wall erosion or at least rock wall weakening is higher in the Randkluft than on a not glacially affected rock. This is however the major question which is not satisfyingly discussed: If I imagine a constant slope and put a glacier at its foot and than I wait for 50'000 years, the rock wall is commonly steeper in the area around the Bergschrund after this time. These steep belts are called Schrundlines and you made a similar observation at your site. Now where does this steepening come from? Is it because the glacier erodes the foot of the rock wall below the Randkluft or does the glacier in contrast protect the rock wall from temperature forcing and decelerates rock erosion while higher erosion rates above the randkluft cause a flattening of the rock wall? Obviously, erosion rates increase after glaciation right? But what is higher? Subglacial erosion or erosion in summit regions not affected by glaciation? Perhaps you can discuss this very basic but unsolved question.

(2) Following your suggestions we added a brief discussion on the implications for (subglacial) headwall retreat. See last paragraph of Sect. 5.1.

(1) I am fundamtally not convinced by these error specification and consider them as much to low and highly theoretical. 1.5 m3 is already the talus which is deposited within failiure scar of a rock fall.

(2) Gaussian error propagation brought the error down to unrealistically low values and was therefore replaced. We updated the error calculation accordingly (which now yields an average relative error of 5.5 %), which is described in Sect. 3.1.2.

**References**

100 Alley RB, Cuffey KM, Zoet LK (2019). Glacial erosion: status and outlook. Annals of Glaciology 60(80), 1–13. https://doi.org/10.1017/aog.2019.38

Evans, IS (2020). Glaciers, rock avalanches and the 'buzzsaw' in cirque development: Why mountain cirques are of mainly glacial origin. Earth Surface Processes and Landforms, DOI: 10.1002/esp.4810.

Sanders, J. W., Cuffey, K. M., Moore, J. R., MacGregor, K. R., and Kavanaugh, J. L.: Periglacial weathering and headwall
105 erosion in cirque glacier bergschrunds, GEOLOGY, 40(9), 779-782, 2012.

Dear Jan Beutel,

thank you for your constructive and insightful comments which we feel helped to improve the manuscript.

We (i) uploaded a revised version of the manuscript (one version with markups showing revisions and responses to minor comments, the other with a clean layout), (ii) posted a new author's comment to inform on general amendments in the manuscript, and (iii) below provide a point-by-point response to your comments. Our response is structured as follows: (1) referee comment, (2) author's response including direct references to manuscript changes.

(1) Specifically I am lacking a pertinent discussion as to the accuracy and validity of your quantitative analysis presented. In no way do I doubt your figures, but in the form they are presented it remains largely unclear how large or small your errors presented are w.r.t. the state of the art and what this errors depend on. I am especially worried since there is no apparent attempt to validate at least part of the figures presented. is it possible to manually cross check the volumes presented with photographs, site visits, deposits on the glacier surface such as seen in fig 4/KNW and KN?

(2) We expanded the description of the error calculation (Sect. 3.1.2) and provided additional information (Sect. 4.2) and figures (Fig. 7 and 8) to validate our observations/analyses.

(1) I am missing a detailed discussion of the exposition and the role of radiation/shading. can you add into figure 5 (polar plot) at which expositions you actually have rock walls in your portfolio and possibly also how much? You show altitude in great detail (fig 6) but little is shown w.r.t. south/north facing. Also your discussion of aggrading/degrading permafrost/active layer is weak and in parts not concise w.r.t. the influence of radiation and the stresses originating from it.

(2) Following your suggestions we added a new subchapter that elaborates on the pattern of rockfall with orientation (see Sect. 4.4 Directional Rockfall Distribution). We furthermore added a paragraph on the sectoral rockfall distribution to the beginning of the discussion (Sect. 5).

(1) In section 5 you discuss that groups of rockfall can be observed e.g. near structural weaknesses or in immediate proximity of the glacier surface. It would be very interesting to see this observation also in your evidence. Can you point out such weaknesses in the topography/photos? Can you point out such hot spots in there as well. Fig. 4 only shows the approximate distribution and size of the observations. But if you are really able to bin these into classes and connect them with properties of the environment, you should really show evidence for that. Maybe only in the form of a spot check and not a total cumulative analysis but without further backing this claim is hard to make. A more detailed discussion and evidence of fracture/weaknesses existing in the Kitzsteinhorn rockwalls would be helpful.

(2) We expanded the study site description (Sect. 2) which now includes information on the dominant discontinuity directions. We furthermore added new content and new figures (Fig. 7 and 8) to Sect. 4.2 to demonstrate the rockfall concentration in weakness zones.

(1) I am not convinced of the discussion of the randkluft as you present it. To my understanding the key property of such deep reaching voids, typically found at the upper boundaries of glacier cirques (not just in the Alps) is that there is no continuous physical contact between ice and ground (rock). This means that there is no mass loading with ice or water pressure and the rock surface is largely exposed to air. So in effect the rock walls are "free standing" compared to vertical (or steep) rock parts that are completely encased in ice. Due to this missing mass loading and the missing water pressure the hydraulic regime changes (see e.g. Simon Loew et al Aletsch Glacier etc.).
Due to the Randkluft reaching deep this is probably the case since a long time - a very long time. Concerning the air and the governing temperature regime I disagree that there is no active layer. It may not be very significant but your

claim about ice cover in mid-October is not convincing, knowing that mid summer is in the end of June and that there is a lot of running water traversing these rock faces from spring (snowmelt) to fall bringing a lot of thermal energy deep into these rock faces below the glacier surface. I rather think that the active layer (and permafrost) regime is of very different properties (temporal, dimensional as well as thermal) as in free surfaces. So maybe you can add thermal data to back up your evidence. In a minimum this should be MAAT, MAGST etc. a discussion of north/south, shaded vs. unshaded etc.

(2) Following your suggestions we added a multi-year temperature dataset from the Randkluft to describe its thermal regime in more detail. Data acquisition and measurement sites are described in Sect. 3.2, results are described in Sect. 4.7, and discussed in Sect. 5.1 and 5.2.

(1) Figure 2: Possibly this figure could be augmented by an even newer picture (end of the study period) to show explicitly the deglaciation that took place during the study period. Also, can you quantify this deglaciation somehow?

(2) We added a new subchapter (Sect. 2.1 Deglaciation) to discuss deglaciation during the study period more explicitly. We furthermore reconstructed the approximate level of the glacier surface from an aerial photo from 1953 (see Fig. 1 and new Fig. 3) and added a new table (Table S8) to the supplement.

(1) Table 2: Rather than repeating the sales brochure of Riegl please specify the settings used for obtaining your data. The general specs of this instruments are known/accessible through the manufacturer to everyone.

(2) Table 2 was removed and information was added to the text. Detailed information on data acquisition parameters are given in Table S2.

(1) Section 3.2: It would help if you give a short synopsis of the algorithms used (M3C2) and not only list benefits similar as how you briefly explain ICP above.

(2) Following your suggestion we provided a synopsis of the algorithm used (new Sect. 3.1.2).

(1) Availability of the data: Is the LIDAR (airborne and terrestrial) available? or can it be made available

(2) The terrestrial LiDAR data is available on request.

(1) Section 4.1: What is the detection limit mentioned? And how is this error determined? can you explain what influences this error (besides the size of the rockfall)? – You mention a theoretical discrepancy w.r.t. the detection. can you please detail here? And besides theory, what does it mean in practice for your study?

(2) We renamed Sect. 4.1. and modified the paragraph. Please also refer to Sect. 3.1.2 where the error calculation is now described in more detail. The error calculation of the M3C2 does not discern between different error sources but instead gives a cumulative error (that factors in all error sources).

(1) Figure 3: What are the two lines? The correlations? Please explain this in detail.

(2) The lines represent the regression lines of the two distributions (all rockfalls; rockfalls > 0.1 m³). We updated the caption.

(1) Section 4.2: Errors of +/- 1.5 m^3 and 1.3m^3 respectively. How sure are you? How did you validate this.

(2) Gaussian error propagation brought the error down to unrealistically low values and was therefore replaced. We updated the error calculation accordingly (which now yields an average relative error of 5.5 %), which is described in Sect. 3.1.2.

(1) Page 12, L 276: You discuss an event that took place before your campaign. If there is a direct context with the observations during your campaign, please explain and back this up with data and plots. If not leave this out. the discussion here is only of a qualitative nature.

(2) The mentioned past event seems to directly control current rockfall activity in the investigated rockwall. Despite the occurrence prior to the start of our monitoring campaign we believe it is justified to mention said event as it helps the reader interpret the observed patterns. We rephrased the relevant passages and hope it is clearer now (Sect. 4.2).

210

(1) Figure 8: Normally CDF functions are given normalized to 100% and not in absolute numbers. Maybe you can also add the thermal data you have to this plot, although clearly one borehole somewhere else is only of limited use in the discussion (you discuss this somewhere later).

(2) We modified the plot. The plot is now given normalized to 100 % and the seasonal maximum of the active layer is indicated.

215

(1) Page 14, L 320: I do not see what hinders debuttressing in this case.

(2) Due to the existence of a Randkluft (air-/snow-filled void) there is no direct contact between the glacier and its headwall (at least down to a certain depth), the glacier does not function as a buttress. Along the no-contact zone we therefore consider debuttressing as irrelevant.

220

**Current glacier recession causes significant rockfall increase:**
**The immediate paraglacial response of deglaciating cirque walls**

[revised manuscript text omitted]

**Kommentiert [IH8]:** Subtitle/subchapter added following suggestion by SC1 (J. Beutel).

We transferred information on deglaciation that before was included in the Study Area section into a separate subchapter and added data on deglaciation trends since the 1950s.

**Kommentiert [IH9]:** Modified following suggestions by RC2 (R. Kenner).

**Kommentiert [IH10]:** Two marginal areas that are not part of the (central) Schmiedingerkees were omitted from the analysis, which reduced the ice loss to 8.5 mio. m³.

**Kommentiert [IH11]:** Modified following suggestion by RC2 (R. Kenner):

RC2 (R. Kenner) wrote:
*"2012 is not that recent..."*

**Kommentiert [IH12]:** This now discussed in more detail in Sect. 3.2 and 4.7

[Figure]

[Figure]

**Figure 1A: Hillshade of study area with monitored rockwalls, scan positions, 1953 glacier extent, and elevation changes of the surface of the Schmiedingerkees glacier between 2008 and 2017. While glacial thinning is most evident near the terminus, pronounced ice surface lowering (~ 0.7 m a⁻¹) is also observed adjacent to the monitored cirque walls. 1B: ft) shows Location of study site within Austria. 1C: Glacier surface lowering 2015-2019. Orange spray marker on rockwall indicates glacier surface level on 04.09.2015 (photo: I. Hartmeyer, 09.09.2019). Abbreviations:  SMK = Scan Position 'Magnetkoepfl', SCC = Scan Position 'Cable Car Top Station', SG1 = Scan Position 'Glacier 1', SG2 = Scan Position 'Glacier 2', SMG = Scan Position 'Maurergrat' (for other abbreviations see text).**

[Figure]

380 **Figure 2: View of Kitzsteinhorn (K) (3.203 m a.s.l.) and Schmiedingerkees glacier (S) from (a) September 1928 (Photo: Stadtarchiv Salzburg, Fotosammlung Josef Kettenhuemer) and (b) September 2011 (Photo: Heinz Kugler). During the reference period the ice surface has lowered considerably while all ice-apronsfaces have completely disappeared. Much of the surface change has occurred since the 1980s. Abbreviations: BN = Borehole North-Face, SCC = Scan Position 'Cable Car Top Station', SMK = Scan Position 'Magnetkoepfl' (for other abbreviations see text).**

385

**Kommentiert [IH13]:** Figure was updated (slight modification/simplification of labelling)

[Figure]

**Figure 3: 2D slope profiles for all monitored rockwalls. Blue arrows indicate approximate level of glacier surface in 1953.**

**Kommentiert [IH14]:** New figure added following suggestion by RC2 (R. Kenner).

RC2 (R. Kenner) wrote:
*"Consider to show some 2D slope profiles including an elevation bar and aspect indication instead of table 1. Decision is up to the authors."*

[Figure]

390

**Figure 4: Geological structure and weakness zones at the monitored rockwalls. 4A: Cleavage (CL) of the calcareous mica-schists dips about 45° NNE. Joint sets J1 (dipping subvertical to W) and J2 (dipping steeply to SW) are approximately orthogonal to CL and predispose north-facing slopes for dip-slope failures; 4B: Highly fractured, slope-parallel escarpment at KN (photo: R. Delleske, 01.08.2018); 4C: Diagonal weakness zone following the direction of cleavage at MKE (photo: R. Delleske, 18.07.2014); 4D: Steep joint sets (J1, J2) predispose east- and west-facing areas to toppling failures (photo: A. Schober, 28.07.2010); 4E: Prominent fault lines resulting from ductile shearing at KNW (photo: R. Delleske, 27.08.2019)**

**Kommentiert [IH15]:** Figure added following suggestions by RC1 (A. Anders) and SC1 (J. Beutel) who criticised missing information on weakness zones and geological structure.

**3 Methods**

**3.1 Terrestrial LiDAR Monitoring**

**3.1.1 Data Acquisition**

Terrestrial LiDAR data acquisition was performed using a Riegl LMS-Z620i laserscanner . A calibrated high-resolution digital camera was mounted on the laserscanner for capturing referenced colour images. Reflectivity on bedrock surfaces was excellent in the near-infrared wavelength used by the scanner, while reflectivity on fresh snow or ice was poor and returned little or no data. Reflectors were not used during data acquisition due to considerable rockfall hazard in the steep, unstable rockwalls.

First LiDAR data was acquired in July/August 2011 at all monitored rockwalls except MKW where data acquisition started in 2012. Data acquisition was restricted to the summer season (May to October).  In total 78 rockwall scans were carried out from five different scan positions. Of these, 22 scans were excluded from further analyses due to snow cover. Scan position 'Maurergrat' was abandoned in 2016, as due to continued glacial thinning site access was lost. Rockwall scans were repeated several times per summer season and at least once per season towards the end of the ablation period. The last scan of all rockwalls was carried out in August 2017, except for MKW that was excluded from further analysis, as unstable blocks were cleared away earlier in 2017 to reduce hazards for a new lift track.

The mean object distances (i.e. distance between scanner and rockwall) differed considerably, varying between 140 m for MKW and 650 m for MGE. The acquisition resolution ranged typically between 0.01-0.02 °,  resulting in point cloud resolution  mostly between 0.1-0.3 m (see Table S3 for full list of data acquisition parameters).

| |  |
|---|---|
|  |  |
|  |  |
|  |  |
|  |  |
|  |  |
|  |  |
|  |  |
|  |  |
|  |  |

**3.1.2 Data Analysis**

Airborne LiDAR datasets acquired in 2008 (Land Salzburg, 2008) were used as base data for georeferencing. Alignment of the acquired sequential point clouds was performed based on surface geometry matching within RiScanPro 1.8. First, point clouds were coarsely registered using the GPS location of the scan position and the azimuth angle of the laserscanner. Numerous techniques exist for the fine registration of point clouds, which include the Iterative-Closest-Point (ICP) algorithm

**Kommentiert [IH16]:** Deleted following suggestion by SC1 (J. Beutel).

SC1 (J. Beutel) wrote:
*"The sentence about rope access is irrelevant for this study. leave it out."*

**Kommentiert [IH17]:** Modified following suggestions by SC1 (J. Beutel).

SC1 (J. Beutel) wrote:
*"Table 2: Rather than repeating the sales brochure of Riegl please specify the settings used for obtaining your data. The general specs of this instruments are known/accessible through the manufacturer to everyone."*

420 (Chen and Medioni, 1992; Besl and McKay, 1992), 3D Least Squares Matching (Akca, 2007), point-to-plane approaches (Grant et al., 2012) and others. Here we used the ICP-algorithm, a popular cloud matching technique for finding the transformation between two point clouds by minimizing the square errors between corresponding entities. Consistent with previous studies on rock slope systems (Rosser et al., 2007; Abellán et al., 2011), alignment errors were  low and  ranged between 1.5-3.7  cm.

425 The two most prominent approaches to identify surface changes in successive point clouds include the identification of homologous objects to calculate displacement fields (Teza et al., 2007; Monserrat and Crosetto, 2008) and direct distance calculation (Rosser et al., 2005). Here, the latter type was applied using the M3C2 algorithm which was specifically designed for orthogonal distance measurement in complex terrain (Lague et al., 2013).

430  M3C2 is frequently used to compute distances between multitemporal point clouds and has been applied in numerous studies investigating geomorphic change (e.g. Barnhart & Crosby, 2013; Cook, 2017; Esposito et al., 2017; James et al., 2017; Williams et al., 2018). Full details can be found in Lague et al. (2013). Briefly, for comparing two successive point clouds A and B, the M3C2 algorithm calculates: (i) a normal vector for any given point $i$ of cloud A by fitting a plane to all neighbouring points $NN_i$ that are within a radius $D/2$ of $i$; (ii) a bounding cylinder of radius $d/2$ with the axis centred at $i$ and oriented

435 normally. Each bounding cylinder isolates subsets of clouds A and B that are projected onto the cylinder axis; (iii) the distribution of distances along the normal is used to calculate mean positions of sub-cloud A ($i_1$) and sub-cloud B ($i_2$). The distance measured ($L_{M3C2}$) between $i_1$ and $i_2$ along the normal direction is stored as an attribute of $i$. The standard deviation of the point distribution within the bounding cylinder (a measure of local roughness) is quantified and combined with the alignment uncertainty to estimate errors and provide a parametric local confidence interval (or level of detection) for each

440 distance measurement. The confidence interval thus represents the sum of different error terms factoring in the cumulative effects of instrumental uncertainty, surface roughness related errors, and alignment uncertainty between point clouds (Hodge, 2010; Soudarissanane et al., 2011). Surface change is considered statistically significant when $L_{M3C2}$ exceeds the local error (confidence interval) and is rejected when $L_{M3C2}$ is smaller than the local error.

Here, a normal scale ($D$) of 5 m was adopted and a projection scale ($d$) of 1.5 m. Plausibility of M3C2 calculations was tested

445 by manually comparing each delineated area of significant surface change (rockfall source area) to computations of the Euclidean nearest-neighbour distance (direct cloud-to-cloud (C2C) calculation). To calculate rockfall volumes, the plausibility-checked results were reanalysed using the M3C2 algorithm, and (i) a fixed normal scale ($D$) (orthogonal to the average local terrain surface) to avoid an overlap of the bounding cylinders and thus an overestimation of rockfall volume, and (ii) using a reduced projection scale ($d$ = 0.25-0.50 m) to avoid integration of unchanged terrain adjacent to the rockfall source

450 area into the distance calculation. Local grids (cell size 5 x 5 cm) containing the $L_{M3C2}$ values of the reanalyses were then created for each rockfall source area and the rockfall volume was computed by grid cell aggregation. The distance measurement error ($L_{M3C2}$ confidence interval) of the grid cells was aggregated for each source area to estimate the rockfall volume error at one sigma level.

**Kommentiert [IH18]:** Modified following suggestions by SC1 (J. Beutel) and RC2 (R. Kenner).

SC1: "*It would help if you give a short synopsis of the algorithms used (M3C2)*"

SC1 and RC1 also both criticised missing information on rockfall volume errors whose calculations is now described in more detail.

455 ~~The M3C2 algorithm has the benefits of: (i) operating directly on point clouds without the need for meshing or gridding, and thus reduces uncertainties; (ii) computing local distances between point clouds along the surface normal direction, which specifically account for terrain roughness; (iii) providing confidence intervals for all distance measurements and thus allowing to assess the significance of surface changes determined; and (iv) providing robust measures on irregular surfaces and with irregularly-spaced data which is important when comparing point clouds of variable resolutions.~~

460  In addition to rockfall volume, following parameters were determined for each source area: mean slope aspect and gradient, elevation above glacier surface as well as maximum depth of rock detachment (determined as the maximum Euclidean nearest-neighbour distance between the pre-event and the post-event point cloud). Source areas were differentiated as bedrock (rockwall) or unconsolidated sediments (intra-rockwall sediment deposits) based on shape,
465 inclination and image colour values. Data gaps due to occlusion are considered negligible for the multitemporal rockwall analysis as obstructions, like deep gullies or protruding spurs that often hamper such analyses in heterogeneous rockwall topography, are rare and scan positions were fixed throughout (except for the final scan at MKW in 2016) for minimising potential detrimental effects from changing incidence angles. Long return periods between surveys however, increase the chance of superimposition and coalescence effects, i.e. adjacent or subsequent events are sampled as one failure only (van
470 Veen et al., 2017; Williams et al., 2018). To improve readability 'rockfall source areas' are referred to as 'rockfalls'.

**3.2 Rockwall Temperature Monitoring**

Bedrock temperature was monitored in two deep and four shallow boreholes. Deep borehole T-BN is located at KN about 40 m above the current glacier surface at 2,985 m a.s.l., and was drilled perpendicular to the ~ 45° terrain surface to a depth of 30 m (Fig. 5). Deep borehole T-BW (25 m) is situated at 2,975 m a.s.l. in a W-facing rock slope (~ 40°) not monitored with
475 terrestrial LiDAR (Fig. 1). Borehole temperature was recorded at eleven (T-BW) and twelve (T-BN) different depths with an accuracy of ± 0.03 °C (Platinum Resistance Temperature Detector L220, 1/10 B, Heraeus Sensor Technology).

A vertical transect consisting of three shallow boreholes (0.8 m deep) was established in a NE-facing section at KN in September 2015 to investigate bedrock temperatures inside the Randkluft (Fig. 5A). The three boreholes are situated (i) at the Randkluft aperture (at glacier surface level) (T-RK1, Fig. 5B and 5E), (ii) 7 m below the glacier surface (T-RK2, Fig. 5C), and
480 (iii) 14 m below the glacier surface (T-RK3, Fig. 5D). Another shallow borehole (0.8 m deep) is located around 5 m above the glacier surface in a ESE-facing section at MKE (T-MKE, Fig. 5A). Temperature in all shallow boreholes is measured with wireless miniature data loggers with an accuracy of ± 0.1 °C (Geoprecision M-Log5W-Rock).

**Kommentiert [IH19]:** Deleted following suggestions by SC1 (J. Beutel).

SC1 (J. Beutel) originally wrote:
*"It would help if you give a short synopsis of the algorithms used (M3C2) and not only list benefits"*

**Kommentiert [IH20]:** SC1 (J. Beutel) criticised that it was unclear how rockfall volume errors are calculated. Explanation of rockfall volume error calculation was therefore expanded (see immediately above).

**Kommentiert [IH21]:** Modified following suggestion by RC2 (R. Kenner):

RC2 (R. Kenner) wrote:
*"How did you considered shadow effects, i.e. data gaps?"*

**Kommentiert [IH22]:** Subchapter added following suggestions by RC1 (A. Anders), SC1 (J. Beutel) and RC2 (R. Kenner) who all asked for a more thorough thermal characterisation of the studied rockwalls.

[Figure]

**Figure 5: Deep and shallow borehole temperature monitoring at Kitzsteinhorn. A: Overview image (photo: R. Delleske, 24.08.2017); B: Measurement site at the Randkluft aperture (T-RK1) (photo: I. Hartmeyer: 04.09.2015); C: Measurement site inside Randkluft, 7 m below glacier surface (photo: R. Delleske, 04.09.2015); D: Measurement site inside Randkluft, 14 m below glacier surface (photo: M. Dörfler, 21.09.2018); E: Close-up of the Randkluft; red dot indicates position of T-RK1 (photo: F. Miesen, 04.09.2016).**

**4 Results**

**4.1 LiDAR Data ResolutionQuality**

To investigate the level of confidence that can be given to the scan results, instrumental and referencing uncertainties were quantified, assumed to be normally distributed, propagated using Gaussian error law and are given at one sigma level. The mean relative error associated with rockfall volumes is 0.1 %. Relative errors are smaller for large rockfall volumes than for small volumes closer to detection limits. Uncertainty for rockfalls smaller than 1 m³ is 0.8 %, while for large rockfalls over 100 m³ relative errors drop to 0.01 % (see Table S3). Data resolution (point density) plays a key role for defining smallest distinguishable detail in point clouds (Hodge, 2010). As a result, more low-magnitude rockfalls will be detected in high-resolution scans compared to low-resolution scans, which introduces issues when rockfall numbers based on scans of differing data resolution are to be compared. The resulting resolution varies between different scans and theoretically leads to the detection of a larger number of small rockfalls in high-resolution scans than in low-resolution scans. This correlation is problematic when scans with different resolutions are compared. To constrain this the influence of data resolution, the mean resulting resolution is compared to the normalizsed number of rockfalls detected (i.e. the number of rockfalls per 10,000 m² per year) in Fig. 6. A, which suggests a weak positive correlation (R² = 0.18) (Fig. 3). can be observed and Ffor rockfalls larger than 0.1 m³ the number of rockfalls is independent of resolution. Alland further analyses werewas limited to this size classrockfall volumes above this volume threshold.. This level of detection Compared to other studies, the minimum usable

**Kommentiert [IH23]:** This is now described in more detail in Sect. 3.1.2

**Kommentiert [IH24]:** Transferred to Sect. 4.2

505 volume of 0.1 m³ derived here is higher is less precise than values specified in related LiDAR-based change detection surveys usingrelying on shorter object distances and higher point densities (e.g. Rosser et al., 2007; Williams et al., 2018) but isis in good agreement with similar monitoring campaigns carried out in high-alpine settings (e.g. Strunden et al., 2015).

Kommentiert [IH25]: Modified following suggestions by SC1 (J. Beutel) and RC2 (R. Kenner).

[Figure]

Figure 63: **Detected number of rockfalls per 10,000 m² a⁻¹ plotted against scan the mean resulting resolution of the performed laserscans. Varying resolutions between acquired scans do not bias the detection of rockfalls larger than 0.1 m³. Dashed lines represent the regression lines of both distributions.**

Kommentiert [IH26]: Modified following suggestion by SC1 (J. Beutel)

**4.2 Inventory of Mass MovementsRockfall Inventory**

During the six-year monitoring period (2011-2017) 632 rockfalls were registered with a total volume of 2,564.3 ± 1.5141.9 m³. When omitting rockfalls below the chosen threshold of 0.1 m³ (Sect. 4.1), the total number drops to 374, while the overall volume is reduced only marginally to 2,551.4 ± 1.3136.8 m³ (Table 31). The mean relative error associated with the rockfall volumes is 5.5 % and similar to other high-alpine LiDAR studies which also found single-digit relative errors (Kenner et al., 2011; Strunden et al., 2015). Relative errors are smaller for large rockfall volumes than for small volumes. Uncertainty for rockfalls smaller than 1 m³ is 29.2 %, while for large rockfalls over 100 m³ relative errors drop to 2.2 % due to reduced cumulative effects of instrumental, surface and alignment errors on larger geometries (Hodge, 2010) (see Table S4).

Large rockfalls over 100 m³ are rare (n = 5) but account for more than two thirds (68.5 %) of the total volume. The largest registered rockfall has a volume of 879.4 ± 6.3 m³, the volumes of the three next largest rockfalls range between 200-300 m³. With increasing volume an exponential decrease in number of rockfalls can be observed. Small rockfalls below 1 m³ represent 80 % of the total number but account for only 3.7 % of the overall rockfall volume (see companion study Hartmeyer et al., 2020) for detailed discussion of magnitude-frequency distributions).

Kommentiert [IH27]: This chapter was expanded following suggestions by SC1 (J. Beutel).

SC1 (J. Beutel) wrote
*"I am especially worried since there is no apparent attempt to validate at least part of the figures presented. is it possible to manually cross check the volumes presented with photographs, site visits, deposits on the glacier surface such as seen in fig 4/KNW and KN?"*

**Table 13: Absolute and normalised rockfall Nnumbers (n) and volumes (m³) of registered rockfalls (> 0.1 
[revised manuscript text omitted]

**Kommentiert [IH29]:** Modified following suggestion by RC2 (R. Kenner).

RC2 (R. Kenner) wrote:
*"How can you determine the volume of a slowly creeping mass?"*

**Kommentiert [IH30]:** Shifted to supplement. (Replaced by new Fig. 7 and 8).

**Kommentiert [IH31]:** Reference to figure added following suggestion by RC2 (R. Kenner).

RC2 (R. Kenner) wrote:
*"not visible at the photo?"*

biggest recorded ice loss was 424.1 ± 60.7 m³ between August 2012 and August 2015 (no data acquisition in 2013 and 2014 due to persistent snow cover), followed by mass losses of 66.8 ± 4.6 m³ and 51.4 ± 4.8 m³ between August 2016 and August 2017, and a mass loss of 33.6 ± 3.8 m³ between August 2015 and August 2016. The two mass losses recorded between 2016 and 2017 are underestimated as late snow cover during the second scan obscured the rockwall.

610  **4.3 Directional  Rockfall Distribution**

Directional analysis of rockwall surface area demonstrates highest shares in the NW-sector (45 %) due to the dominant influence of KNW, followed by E- and N-facing areas (17 % each). S-oriented source areas are largely missing due to the absence of significant S-faces in the study area (Fig. 9A). The bulk of the registered rockfalls originated in N- and NW-oriented rockwall sections between 2,900 and 3,100 m a.s.l.
615  . Est-facing rockfall source areas between 2,800 and 3,000 m a.s.l. represent another distinct azimuth cluster (Fig. 9B). Normalised rockfall volume (i.e. rockfall volume per 10,000 m² per year) is significantly elevated in the N-, W- and SE-sector and low for all other slope aspect sectors (Fig. 9C). Normalised rockfall numbers (i.e. rockfall number per 10,000 m² per year) peak in the same sectors as normalised rockfall volume (except for the S-sector, which however is represented by just a single rockfall) (Fig. 9D). High (normalised) activity in N-, W- and SE-sectors most likely
620  reflects the dominant discontinuity orientations at the study site which promote the detachment of cubic to rhomboidal detachments in N-facing areas and failures along steep joint systems in W- and SE-facing areas (Fig. 4).

**Kommentiert [IH32]:** This subchapter was added following suggestions by RC1 (A. Anders) and SC1 (J. Beutel) to provide more information on the directional/sectoral distribution of rockfalls.

[Figure]

**Kommentiert [IH33]:** Figure modified (see new Fig. 9) following suggestions by RC1 (A. Anders), SC1 (J. Beutel) and RC2 (R. Kenner) who asked for more information on the sectoral distribution of rockfalls.

[Figure]

**Figure 9A: Sectoral distribution of surface area. 9B: Slope azimuth angle and elevation above sea level for all rockfalls. 9C: Normalised rockfall volume by sector (m³ 10,000 m⁻² a⁻¹). 9D: Normalised rockfall number by sector (n 10,000 m⁻² a⁻¹).**

**4.5 Altitudinal Rockfall Distribution**

To further explore rockfall distribution with elevation, we classified the investigated surfaces into vertical 50 m bins. Total rockwall surface area is almost normally distributed and shows the largest share between 2,900-2,950 m a.s.l. (~ 65,000 m²; 27.0 % of total surface area) (Fig. 610a and Table S65). Normaliszed and absolute rockfall volumes peak in the same elevation class. Over 37 m³ per 10,000 m² a⁻¹ originated between 2,900-2,950 m a.s.l. (Fig. 610b), which is equivalent to more than half (54.4 %) of the total rockfall volume (Table S65).

The normaliszed number of rockfalls is highest between 2,950-3,000 m a.s.l. (4.6 rockfalls per 10,000 m² a⁻¹) and declines significantly . Wwith increasing/decreasing elevation, normalized rockfall numbers decline significantly, similar to the altitudinal distribution of surface area (Fig. 610c). Absolute rockfall numbers peak between 2,900-2,950 m a.s.l. and 2,950-3,000 m a.s.l., where approximately two thirds (63.8 %) of the detected rockfalls originate (Table S65).

To detail the vertical distribution of rockfall source areas, the elevation differences between rockfall source areas and local glacier surface are calculated and grouped into 10 m bins (Fig. 711a and Table S21). Immediately above the glacier surface (0-10 m) rockfall volumes are by far the highest (75.6 m³ per 10,000 m² a⁻¹) (Fig. 711b). 60 % of the total rockfall volume detached from this segment, which constitutes only 15 % of the total rockwall surface area. With increasing distance from the glacier surface, a sharp decrease in rockfall volume can be observed. In the next higher segment (10-20 m), normaliszed rockfall volume slightly exceeds 20 m³ per 10,000 m² a⁻¹, while in all other height classes rates remain below 10 m³ per 10,000 m² a⁻¹. Only in two segments (90-100 m, 170-180 m), this pattern is masked by the presence of comparatively large, singular rockfalls.

> **Kommentiert [IH34]:** Modified following correction by RC2 (R. Kenner).
>
> RC2 (R. Kenner) wrote:
> *"Is this of relevance if the values are normalized by area?"*

[Figure]

**Figure 106: (a) Rockwall surface area, (b) normaliszed rockfall volume and (c) the normaliszed number of rockfalls, grouped by elevation above sea level. Between 2,900 and 2,950 m a.s.l. a distinct peak in rockfall volume is observed.**

Analysed individually, a positive correlation between rockfall volume and proximity to glacier surface occurs for all rockwalls except KNW. The vast majority of the rockfall volume is detected within 10 m of the glacier surface at MGE (73 %), KN (79 %) and MKE (98 %). Considering the first 20 m above glacier surface the volume percentages exceed 90 % for all three rockwalls.

660 At MKW rockfall volumes are small in the lowest segment (3 %) and 96 % of the  rockfall volume occur in the segment above (10-20 m). Here, rockfall activity in decades prior to the start of the monitoring, created a steep scarp around 15 m above the current glacier surface. The rockfall deposits, likely several thousand cubic meters of rock, accumulated at the foot of MKW and constituted a talus cone that decoupled parts of the rockwall from the glacier. Numerous rockfalls detached
665 from this scarp during the study period, indicating continued stress release after the  preceding events.

As mentioned no pronounced glacial proximity pattern was found for KNW, where only 12 % of the rockfall volume detached within the first 10 m. Here, a significant 272.7 ± 11.4 m³ rockfall occurred in summer 2016 (Sect. 4.2) which constituted
670 around half of the total rockfall volume at this site.  Still, after excluding this event, only a rather weak proximity pattern is observed (23 % of the volume within the lowest 10 m) clearly deviating from the patterns observed at the other four rockwalls.

Analysis of rockfall numbers confirms the glacial proximity pattern even though the correlation is much less pronounced than for the elevation volume distribution. Highest normalised rockfall numbers (3.9 rockfalls per 10,000 m² a⁻¹) are once again
675 found in the lowest segment (0-10 m) (Fig. 7c). The mean value for all higher segments (i.e. 10-260 m) equals 2.5 rockfalls per 10,000 m² a⁻¹ with significant variations between the different height classes. Overall 21 % of all rockfalls (78 of 374) occurred in the first 10 m above the glacier surface – a distinct contrast to the dominance of rockfall volumes in that segment. Comparing rockfall numbers across the rockwalls yields diverse results: At KN particularly high rockfall numbers are found between 30 and 50 m above the glacier. KNW shows a more uniform pattern with a rather balanced distribution over the first
680 100 m and a slight decrease at higher elevations. At MKE, rockfall is restricted to the immediate adjacency to the glacier and above the 0-10m-segment only minimal rockfall activity is observed. At MKW and MGE, most rockfalls occurred within 20 m of the glacier surface (~ 70 % and 90 %, respectively).

**Kommentiert [IH35]:** Modified following question by RC2 (R. Kenner).

RC2 (R. Kenner) wrote:
*"this is ambigious. total rock fall volume or mean rock fall volume?"*

**Kommentiert [IH36]:** Slightly modified for consistency with new passages in Sect. 4.2.

**Kommentiert [IH37]:** Modified following question by RC2 (R. Kenner):

RC2 (R. Kenner) wrote:
*"This is impossible when looking at figure 7c?"*

Explanation: Fig. 7c (now Fig. 11c) gives the normalised rockfall number.

[Figure]

**Figure 117: (a) Rockwall surface area, (b) normalissed rockfall volume and (c) the normalizsed number of rockfalls, classified by elevation above glacier surface. Areas exposed by recent glacier retreat are heavily susceptible to rockfall, during the observation period (2011-2017) 60 % of the total rockfall volume detached within 10 m of the current glacier surface.**

The observed distribution of rockfall magnitudes and frequencies is described by a distinct negative power function over four orders of magnitude (Fig. 12). To test the statistical robustness of the discovered differences between glacier-proximal (< 10 m above glacier) and glacier-distal (> 10 m above glacier) rockfall activity the goodness of fit was analysed using a bootstrapping approach (full details are given in Hartmeyer et al., 2020). In the analysis 20 % of the rockfalls were randomly removed and the data set was resampled 100,000 times to assess the sensitivity of the power-law-exponent $b$, which represents a frequently used variable to characterise spatiotemporal rockfall variation (e.g. Dussauge-Peisser et al., 2002; Barlow et al., 2012). Results demonstrate robust power law exponent $b$ estimates of $0.51^{+0.07}_{-0.05}$ for the proximal and $0.69^{+0.04}_{-0.03}$ for the distal datasets at 95 % confidence level.

To selectively examine the statistical sensitivity to individual rare events, the power-law-fits were recalculated after omitting the five largest rockfalls (volumes > 100 m³). Power law exponents for proximal rockfalls ($0.59^{+0.07}_{-0.05}$) and distal rockfalls ($0.71^{+0.04}_{-0.03}$) only slightly increase and show the significance of the differences observed between proximal and distal areas. Normalised rockfall volume in proximal areas (11.7 m³ per 10,000 m² a⁻¹) was 2.6 times higher than in distal areas (5.2 m³ per 10,000 m² a⁻¹) in the reduced data set (volumes > 100 m³ omitted).

**Kommentiert [IH38]:** Modified following suggestion by RC2 (R. Kenner)

RC2 (R. Kenner) wrote
*"You did a good job in estimating the accuracy of your measurements. However, I miss something similar for the statistics. Some of the statistical analysis are based on very small sample sizes or are strongly influenced by single events. I wonder if all this is significant and would appreciate something like a sensitivity analysis."*

[Figure]

700 **Figure 12: Magnitude-frequency distributions for glacier-proximal rockfalls and glacier-distal rockfalls. Proximal rockfalls (b = 0.51) are fitted by a significantly flatter regression line than distal rockfalls (b = 0.69) indicating an increased occurrence of large events in recently deglaciated areas.**

**4.6 Rockfall Failure Depths**

705 Among the 374 rockfalls identified, depth of failure  ranges between 0.17 and 6.45 m. Near-surface failures  dominane as 69 % of all rockfalls failed within the top 0.5 m and another 22 % in depths between 0.5 and 1 m. Eleven rockfalls with failure depths of more than 2.0 m were recorded (2.9 %) and only five rockfalls failed in depths larger than 3.0 m (1.3 %) (Fig. 13, Table S6). Classification of rockfall failure depth by slope aspect demonstrates an increased occurrence of relatively deep failures (> 1 m, > 2 m) in W-,

710 SE-, and N-facing areas (Table 2). This pattern is consistent with dominant local discontinuities which predispose N-facing rockwall sections to thick dip-slope failures along the NNE-dipping cleavage, as well as W- and SE-facing sections to large (toppling) failures along the steep joint sets J1 and J2 (Fig. 4).

> **Kommentiert [IH39]:** Modified following suggestion by RC1 (A. Anders).
>
> RC1 (A. Anders) wrote:
> *"Is there a difference in the thickness of the rockfalls with orientation (or a pattern related to location)?"*

[Figure]

**Figure 138: Failure depths  for all registered rockfalls (n = 374). More than 90 % of all rockfalls failed within less than 1 m from the surface. The seasonal active layer maximum measured at T-BN (30 m deep borehole at KN) ranged from 3.0-4.2 m (2016-2019).**

**Kommentiert [IH40]:** Modified following suggestion by SC1 (J. Beutel).

SC1 (J. Beutel) wrote:
*"Normally CDF functions are given normalized to 100% and not in absolute numbers. Maybe you can also add the thermal data you have to this plot, although clearly one borehole somewhere else is only of limited use in the discussion (you discuss this somewhere later)."*

[revised manuscript text omitted]

**Kommentiert [IH50]:** Modified following suggestion by RC2 (R. Kenner).

RC2 (R. Kenner) wrote:
*"You should mention the infiltration of water after AL formation what is one of the key game changers."*

**Kommentiert [IH51]:** This paragraph was expanded/modified to briefly discuss the newly included temperature data.

Further influences that potentially contribute to high glacier-proximal rockfall activity, include late-spring ground avalanches and channelized rainwater runoff after heavy precipitation. Visual observation suggests strong erosive effects for these processes in the freshly deglaciated sections where blocks at failure stability limit are abundant (Fig. 16B). More precise quantification of such processes would require significantly shorter survey return periods.

[Figure]

**Kommentiert [IH52]:** Old Fig. 9 and 10 were merged into new Fig. 16A and 16B

**6 Conclusions**

We present a unique rockfall inventory from a six-year terrestrial LiDAR campaign (2011-2017) for permafrost-affected rockwalls of two glaciated cirques in the Central Alps of Austria (Kitzsteinhorn). The five rockwalls studied are all influenced by significant glacial downwasting and ice-apronface degradation. We draw the following conclusions:

- The inventory represents the most extensive dataset of high-alpine rockfall to date and the first quantitative documentation of a cirque-wide erosional response of glaciated rockwalls to recent climate warming.

- During the monitoring period 632 rockfalls with an overall volume of 2,564.3 ± 1.5141.9 m³ were recorded. In addition, 113 rockfall source areas with a total volume of 292.0 ± 72.30.4 m³ were detected in unconsolidated sediments. Mass loss from ice-apronface degradation accounted for an overall volume of 575.9 ± 0.04 73.9 m³.

- Rockfall activity concentrates along pre-existing structural weaknesses and was highest in recently deglaciated areas: 60 % of the rockfall volume originated from source areas located fewer than ten vertical meters above the current glacier surface; 75 % detached within 20 vertical meters of the glacier surface.

- Increased mass wasting activity in recently deglaciated areas, such as discovered in the present study, is typical of paraglacial environments, where slope systems gravitationally adjust to new, non-glacial boundary conditions.

- Previous studies on the paraglacial adjustment of bedrock slopes mostly focused on high-magnitude events such as rock avalanches and rockslides, which commonly respond to deglaciation on centennial to millennial time scales. The lower end of the paraglacial magnitude-frequency spectrum is currently poorly characterized. The present study bridges this gap and for the first time provides field evidence of an immediate, low-magnitude paraglacial response in a currently deglaciating rock slope system.

- Distinct Randklufts, which separate the investigated cirque walls from the adjacent glacial ice, effectively prevent debuttressing. To characterise the thermal regime of the Randkluft we carried out unprecedented shallow borehole (0.8 m) measurements at 7 and 14 m Randkluft depth, and found Inside the Randkluft we observed perennially frozen conditions and extensive refreezing of meltwater supplied from the rockwall above.

- Sustained freezing along with sufficient water availability in the Randkluft likely drive subcritical stress propagation by ice segregation and cause high pluckingquarrying-related tensile stresses, which contribute to antecedent rockfall preparation when the rockwall is still ice-covered.

**Kommentiert [IH53]:** Modified following suggestion of RC2 (R. Kenner) to more explicitly consider the role ice segregation.

- As the glacier is wasting down strong diurnal and seasonal temperature variations induce pronounced thermal stress, cause rock fatigue and lead to the first-time formation of an active layer, which is expected to exert a significant destabilizing effect on glacier-proximal areas.

**Data availability.** The rockfall inventory can be downloaded from the *mediaTUM* data repository under the following weblink: https://mediatum.ub.tum.de/1540134. Terrestrial LiDAR data is available on request.

**Kommentiert [IH54]:** Modified following suggestion by SC1 (J. Beutel)

SC1 (J. Beutel) wrote:
*"Is the LIDAR (airborne and terrestrial) available? or can it be made available`?"*

[revised manuscript text omitted]

---

## Editor Decision (ED1)

[revised manuscript text omitted]

*Reference periods for MKW: 1953-2016 and 2012-2016.

45

[Figure]

**Figure S1: Rockfall source areas and volumes for all five monitored rockwalls (Photos: Robert Delleske).**

**Figure S2: Images from the investigated Randkluft below the Kitzsteinhorn north-face (KN). A: Picture was taken at ~ 10 m depth, permanently ice-covered headwall is on the left, glacier is on the right (Photo: I. Hartmeyer, 09.10.2015). B: Randkluft is narrowing with depth. Picture was taken at ~ 15 m depth, T-RK3 is visible in the background (headwall is on the right, glacier on the left) (Photo: I. Hartmeyer, 13.09.2019). C: Melt- and rainwater entering the Randkluft contributes to the formation of thick ice coatings and icicles on the rockwall (Photo: I. Hartmeyer, 16.10.2018).**

50

[Figure]

55

**Figure S3: Temperature record from 25 m deep borehole T-BW (2,975 m a.s.l., Kitzsteinhorn W-face). Temperature at 25 m depth ranged between -1.1 and -1.2 °C during the observation period. Repeated lightning  damage caused fragmentary data availability.**

---

## Author Response (AR2)

Dear Editors,

thank you very much for your detailed and informative feedback. We implemented all requested technical corrections.
Below you find a marked-up manuscript version.

Kind regards
Ingo Hartmeyer (on behalf of all co-authors)

[revised manuscript text omitted]

**Kommentiert [IH3]:** AE A. Stroeven wrote:
"is this the width?"

Response:
B (= sensor class) describes the accuracy of the sensor. Further info: https://www.thermometricscorp.com/acstan.html

"1/10 B" is removed because the accuracy was already specified (+/- 0.03 °C).

**Kommentiert [IH4]:** AE A. Stroeven wrote:
*"sure? If I Google "Geoprecision M-Log5W-Rock" I find 0.25 C?"*

Response:
At the Geoprecision website the specified accuracy is +/- 0.1 °C (Link: http://www.geoprecision.com/produkte-d/funk-datenlogger-433mhz)

(Note: It is easy to confuse the M-Log5W-**Rock** with the M-Log-5W-**Dallas**, which in fact has an accuracy of +/- 0.25 °C).

**Figure 5:** Deep and shallow borehole temperature monitoring at Kitzsteinhorn. **5**A:  Air photo (photo: R. Delleske, 24.08.2017); **5**B: Measurement site at the Randkluft aperture (T-RK1) (photo: I. Hartmeyer: 04.09.2015); **5**C: Measurement site inside Randkluft, 7 m **below** the 2015 glacier surface (T-RK2) (photo: R. Delleske, 04.09.2015); **5**D: Measurement site inside Randkluft, 14 m below glacier surface (T-RK3) (photo: M. Dörfler, 21.09.2018); **5**E: View of the investigated Randkluft from the cable car top station (CC); red dot indicates position of T-RK1 (photo: F. Miesen, 04.09.2016).

265

**Kommentiert [IH5]:** Labelling updated: "CC" added to 5A.

**Kommentiert [IH6]:** AE A. Stroeven wrote:
"are values below the glacier surface consistently in reference to 2015? This should be mentioned."

Response:
Yes, all values consistently refer to the glacier surface level in September 2015. Info was added to the caption and the text.

**4 Results**

**4.1 LiDAR ata esolution**

[revised manuscript text omitted]

**Kommentiert [IH8]:** Fig. 8B was updated.

**Kommentiert [IH9]:** AE A. Stroeven wrote:
"hard to grapple for me: just to make sure, this is not the post-event photo (i.e. have they been switched?)"

Response:
No, they haven't been switched. 8A really is the pre-event photo, while 8B is the post-event photo. But I absolutely admit that the photos are a little hard to interpret without knowledge of the terrain (it's hard to visualize joint bodies on frontal photos). I expanded the caption of 8A and 8B and hope it is clearer now.

(I also tried to mark the visible discontinuities (CL, J2) but that clutters/confuses the figures a little bit.)

[revised manuscript text omitted]

Kommentiert [IH12]: AE A. Stroeven wrote:
*"isn't this the first mention of this number? perhaps remove, it is not part of the abstract either..."*

Response:
Excellent point. We added the information to Sect. 4.5 (text & caption of Fig. 11)

**Author contributions.** MKE, LS and JO initiated the underlying research project in 2010 and obtained the funding. IH, MKE and RD developed the idea and designed the study. IH and RD conducted the data acquisition and IH analysed the data. All authors contributed to the discussion and interpretation of the data. IH drafted the manuscript with significant contributions from MKR and AL.

**Competing interests.** The authors declare that they have no conflict of interest.

**Acknowledgements**. We would like to thank Alison Anders, Jan Beutel, Robert Kenner, and Associate Editor Arjen Stroeven for their thoughtful feedback and constructive reviews. A. Schober, H. Kugler, A. Schober, M. Dörfler, and F. Miesen kindly provided the photographs used for Fig. 3D2B, 4D4B, 5D, and 5E.

**Financial support.** This study was co-funded by the Austrian Academy of Sciences (ÖAW) (Project 'GlacierRocks'), the Arbeitsgemeinschaft Alpenländer (ARGE ALP) (Project 'CirqueMonHT') and the Austrian Research Promotion Agency (FFG) (Project 'MOREXPERT'). We furthermore thank the Gletscherbahnen Kaprun AG (Project 'Open-Air-Lab Kitzsteinhorn') for financial and logistical support.

[revised manuscript text omitted]

*Reference periods for MKW: 1953-2016 and 2012-2016.

**Table S3: Key data acquisition parameters for each of the five monitored rockwalls. Data acquisition was performed with a Riegl LMS-Z620i laserscanner (accuracy ± 10 mm (1σ @ 100 m)).**

[revised manuscript text omitted]

**Failure Depth (m)**

| | | < 0.5 | 0.5 – 1.0 | 1.0 – 1.5 | 1.5 – 2.0 | 2.0 – 2.5 | 2.5 – 3.0 | > 3.0 | Total |
|---|---|---|---|---|---|---|---|---|---|
| **TOTAL** | Nr. (n) | 258 | 83 | 20 | 2 | 5 | 1 | 5 | 374 |
| | % | 69.0 | 22.2 | 5.3 | 0.5 | 1.3 | 0.3 | 1.3 | 100.0 |
| **KN** | Nr. (n) | 73 | 17 | 9 | 2 | - | - | 3 | 104 |
| | % | 70.2 | 16.3 | 9.7 | 1.9 | - | - | 2.9 | 100.0 |
| **KNW** | Nr. (n) | 145 | 29 | 3 | - | 2 | - | - | 179 |
| | % | 81.0 | 16.2 | 1.7 | - | 1.1 | - | - | 100.0 |
| **MKE** | Nr. (n) | 9 | 7 | 5 | - | 1 | - | 1 | 23 |
| | % | 39.1 | 30.4 | 21.7 | - | 4.3 | - | 4.3 | 100.0 |
| **MKW** | Nr. (n) | 6 | 5 | 2 | - | 1 | 1 | - | 15 |
| | % | 40.0 | 33.3 | 13.3 | - | 6.7 | 6.7 | - | 100.0 |
| **MGE** | Nr. (n) | 25 | 25 | 1 | - | 1 | - | 1 | 53 |
| | % | 47.2 | 47.2 | 1.9 | - | 1.9 | - | 1.9 | 100.0 |

970

| |  |  |  |  |  |  |
|---|---|---|---|---|---|---|
| |  |  |  |  |  |  |
|  | | | | | | |
|  |  |  |  |  |  |  |
|  |  |  |  |  |  |  |
|  | | | | | | |
|  |  |  |  |  |  |  |
|  |  |  |  |  |  |  |

975

[Figure]

**Figure S17: Rockfall source areas and volumes for all five monitored rockwalls (Photos: Robert Delleske).**

[Figure]

Figure S2: Images from the investigated Randkluft below the Kitzsteinhorn north-face (KN). A: Picture was taken at ~ 10 m depth, permanently ice-covered headwall is on the left, glacier is on the right (Photo: I. Hartmeyer, 09.10.2015). B: Randkluft is narrowing with depth. Picture was taken at ~ 15 m depth, T-RK3 is visible in the background (headwall is on the right, glacier on the left) (Photo: I. Hartmeyer, 13.09.2019). C: Melt- and rainwater entering the Randkluft contributes to the formation of thick ice coatings and icicles on the rockwall (Photo: I. Hartmeyer, 16.10.2018).

980

[Figure]

985

**Figure S18:** Temperature record from 25 m deep borehole T-BW (2,975 m a.s.l., Kitzsteinhorn W-face). Temperature at 25 m depth ranged between -1.1 and -1.2 °C during the observation period. Repeated lightning stroike damage caused fragmentary data availability.

[Figure]

990  **Figure S19: Images from the investigated Randkluft below the Kitzsteinhorn north-face (KN). A: Picture was taken at ~ 10 m depth, permanently ice-covered headwall is on the left, glacier is on the right (Photo: I. Hartmeyer, 09.10.2015). B: Randkluft is narrowing with depth. Picture was taken at ~ 15 m depth, T-RK3 is visible in the background (headwall is on the right, glacier on the left) (Photo: I. Hartmeyer, 13.09.2019). C: Melt- and rainwater entering the Randkluft contributes to the formation of thick ice coatings and icicles on the rockwall (Photo: I. Hartmeyer, 16.10.2018).**